# Cohesive and mixed sediment in the Regional Ocean Modeling System (ROMS v3.6) implemented in the Coupled Ocean Atmosphere Wave Sediment-Transport Modeling System (COAWST r1179)

Christopher R. Sherwood[1], Alfredo L. Aretxabaleta[1], Courtney K. Harris[2], J. Paul Rinehimer[2,3], Romaric Verney[4], Bénédicte Ferré[1, 5]

[1]U. S. Geological Survey, 384 Woods Hole Road, Woods Hole, MA 02543-1598 USA
[2]Virginia Institute of Marine Sciences, Gloucester Point, Virginia, USA
[3]Currently at WEST Consultants, Bellevue, WA, USA
[4]IFREMER, Plouzane, France
[5]Currently at CAGE-Centre for Arctic Gas Hydrate, Environment, and Climate; Department of Geosciences, UiT The Arctic University of Norway, N-9037 Tromsø, Norway

*Correspondence to*: Christopher R. Sherwood (csherwood@usgs.gov)

**Abstract.** We describe and demonstrate algorithms for treating cohesive and mixed sediment that have been added to the Regional Ocean Modeling System (ROMS version 3.6), as implemented in the Coupled Ocean Atmosphere Wave Sediment-Transport Modeling System (COAWST Subversion repository revision 1179). These include: floc dynamics (aggregation and disaggregation in the water column); changes in floc characteristics in the seabed; erosion and deposition of cohesive and mixed (combination of cohesive and non-cohesive) sediment; and biodiffusive mixing of bed sediment. These routines supplement existing non-cohesive sediment modules, thereby increasing our ability to model fine-grained and mixed-sediment environments. Additionally, we describe changes to the sediment bed-layering scheme that improve the fidelity of the modeled stratigraphic record. Finally, we provide examples of these modules implemented in idealized test cases and a realistic application.

**Copyright statement**

# 1 Introduction

## 1.1 Motivation

Fine cohesive sediment (mud) is present in almost every coastal environment, and influences water clarity, benthic habitats, shoaling of harbors and channels, storage and transport of nutrients and contaminants, and morphologic evolution of wetlands, deltas, estuaries, and muddy continental shelves (Winterwerp and van Kesteren, 2004; Edmonds and Slingerland, 2010; Caldwell and Edmonds, 2014; Mehta, 2014; Li et al., 2017). The properties and behavior of mud depend on more than the size, shape, and density of the individual particles, so they are more difficult to characterize and model than properties of non-cohesive material like sand. Cohesive sediment often forms flocs that have lower densities, larger diameters, and faster settling velocities than the primary particles. Acoustic and optical sensors respond differently to suspensions of flocculated sediment, compared with similar mass concentrations of unflocculated particles, and these responses have important influences on observations of suspended-sediment mass concentrations, especially in estuaries (for example, McCave and Swift, 1976; McCave, 1984; Eisma, 1986; Hill and Nowell, 1995; Winterwerp, 1999, 2002; Winterwerp et al., 2006; Xu, Wang, and Riemer, 2008; 2010; Verney et al., 2011; Slade, Boss, and Russo, 2011; MacDonald et al., 2013; Thorne et al., 2014).

Cohesive sediment beds are distinguished by generally finer sediment, including some clay content, often are poorly sorted, and have low bulk density (high water content). Cohesive beds have a tendency for bulk responses to bottom stress, rather than individual particle responses. Cohesive beds have rheological properties that can range from fluids to Bingham plastics to granular materials, and may change with time in response to changes in water content, biochemical processes and fluid or geomechanical stresses (Dyer, 1986; Whitehouse et al., 2000; Winterwerp and Kranenburg, 2002; Winterwerp and van Kesteren, 2004; Maa et al., 2007; Knoch and Malcherek, 2011; Mehta, 2014).

Sediment transport in coastal ocean models is sensitive to the representation of fine-scale stratigraphy because evolving seabed properties determine what sediment is exposed to the water column and available for transport. Small-scale stratigraphy and grain-size distribution at the sediment-water interface also influence the grain roughness of the seabed, affect the type of small-scale roughness (biogenic features and ripples) present on the bed, and control properties like acoustic impedance of the seafloor. Biodiffusion influences stratigraphy by reducing gradients in grain size and other bed properties and by mixing materials from deeper in the bed to closer to the surface, where they may be more susceptible to transport.

## 1.2 Previous Modeling Efforts

Amoudry and Souza (2011) surveyed regional-scale sediment-transport and morphology models, and found that one of the shortcomings was the treatment of cohesive- and mixed-sediment models. The water-column behavior of cohesive sediment (e.g., flocculation and disaggregation, and settling) and the consolidation of settling particles to form a cohesive bed has been modeled mostly with one-dimensional vertical (1DV) models or with empirical formulae that allow particle settling velocity to vary as a function of salinity (Ralston et al., 2012) or suspended-sediment concentration (e.g., Mehta, 1986; Lick et al., 1993; Van Leussen, 1994; Lumborg and Windelin, 2003; Lumborg, 2005; and Lumborg and Pejrup, 2005). Mietta et al. (2009) have demonstrated the effect that pH and organic-matter content have on mean floc size and settling velocity. The primary dynamical effect of flocculation is to increase settling velocities, thereby increasing the mass settling flux. Soulsby et al. (2013) reviewed methods for estimating floc settling velocities and proposed a new formulation that depends primarily on turbulence shear and instantaneous suspended-sediment concentration. Spearman et al. (2011) noted that adjustments to settling velocity (e.g., Manning and Dyer, 2007) were able to successfully reproduce floc settling in one-dimensional estuary modeling applications. However, approaches that adjust only settling velocity do not allow analysis of other characteristics of the suspended particle field, such as particle size and density, which affect acoustic and optical properties, or geochemical properties (water content and surface area). Full floc dynamics have been incorporated in only a few coastal hydrodynamics and sediment-transport models. Winterwerp (2002) incorporated his floc model (Winterwerp, 1999) in a three-dimensional simulation of the estuary turbidity maximum (ETM) in the Ems estuary. Ditschke and Markofsky (2008) described formulations in TELEMAC-3D to represent exchanges among size classes from floc dynamics. Xu et al. (2010) added floc dynamics to the Princeton Ocean Model (POM) and simulated the ETM in Chesapeake Bay. Empirical formulae for the erosion of cohesive sediment have been derived from laboratory flume measurements and field experiments (Whitehouse et al, 2000; Mehta, 2014). Many have a form similar to the Ariathurai and Arulanandan (1978) equation used in ROMS (Warner et. al., 2008), which relates erosional flux $E$ (kg m$^{-2}$ s$^{-1}$) to the normalized excess shear stress as $E = E_0 (1 - \phi) \left[ (\tau_{sf} - \tau_c) / \tau_{sf} \right]$ when $\tau_{sf} > \tau_c$, and where $E_0$ (kg m$^{-2}$ s$^{-1}$) is an empirical rate constant, $\phi$ (m$^3$/m$^3$) is sediment porosity, $\tau_{sf}$ (Pa) is the skin-friction component of the bottom shear stress, and $\tau_c$ (Pa) is the critical shear stress for erosion. Erosion of cohesive sediment in some models (for example Delft3D; van der Wegen et al., 2011; Caldwell and Edmonds, 2014) uses a similar formulation, subject to a user-specified critical shear stress for erosion. It is recognized that $\tau_c$ may increase with depth in sediment, and erosion-rate formulae have been proposed that incorporate depth-dependent profiles for $E_0$ and/or $\tau_c$ (Whitehouse et al, 2000; Mehta, 2014). Wiberg et al. (1994) demonstrated the need to account for small-scale stratigraphy to represent bed armoring for a non-cohesive model, and did so via a layered bed model that kept track of changes to sediment-bed grain-size distribution in response to cycles of erosion and deposition. Bed layers have been used to

represent temporal changes to bed erodibility for fine-grained sediment, for example by using an age model for the bed
(HydroQual, 2004). Biodiffusion may alter stratigraphy, and there are many 1DV models that treat diffusive mass flux of
sediment and reactive constituents in the bed, mostly motivated by water-quality and geochemical concerns (e.g., Boudreau,
1997; DiToro, 2001; Winterwerp and van Kesteren, 2004). Several regional-scale circulation and sediment-transport models
treat sediment stratigraphy, including ECOMSED (HydroQual, 2004), ROMS/CSTMS (Warner et al., 2008), Delft3D (van
der Wegen et al., 2011), FVCOM, TELEMAC/SISYPHE (Villaret et al., 2007; Tassi and Villaret, 2014), MARS3D (Le Hir,
2011; Mengual et al., 2017) and some have unpublished treatments for cohesive processes. Sanford (2008) pioneered an
approach where the critical shear stress for each bed layer was nudged toward an assumed equilibrium value, and the critical
stress for erosion of the surface layer alternately became smaller or larger in response to deposition and erosion. We have
combined the approach of Sanford et al. (2008) with biodiffusive mixing to represent depth-dependent changes in erodibility.
This approach has been implemented in the cohesive bed stratigraphy algorithm in ROMS (described here) and applied by
Rinehimer et al. (2008), Butman et al. (2014), and Fall et al. (2014).

**1.3 Goals of the Model**

Our goal in developing and refining sediment dynamics in ROMS is to produce an open-source community model
framework useful for research and management that combines cohesive and non-cohesive behavior and is suitable for
simulating sediment transport, stratigraphic evolution, and morphologic change. Our goal is to develop methods that can be
implemented within coastal and estuarine models for application at regional scales, i.e. domains of 10s to 100s of km$^2$ with
grid elements of $10 - 10,000$ m$^2$ and the ability to resolve time scales ranging from minutes to decades.

**1.4 Objectives and Outline of the Paper**

The behavior of non-cohesive sediment (sand) in ROMS was described by Warner et al. (2008). ROMS also includes several
biogeochemical modules (Fasham et al., 1990; Fennel et al., 2006). New components have since been added, including
spectral irradiance and seagrass growth models (del Barrio et al., 2014) and a model for treating the effects of submerged
aquatic vegetation on waves and currents (Beudin et al., 2017). The present paper describes new components that model
processes associated with cohesive sediment (mud) and mixtures of sand and mud. These include aggregation and
disaggregation of flocs in the water column, sediment exchange with a cohesive bed where erosion is limited by a bulk
critical shear stress parameter that increases with burial depth, and tracking stratigraphic changes in response to deposition,
erosion, and biodiffusive mixing. Our goal is to demonstrate that the algorithms reproduce some of the important behaviors
that distinguish cohesive sedimentary environments from sandy ones, and to demonstrate their utility for modeling muddy
environments. The model processes are presented and discussed in Section 2. Additional details of the model implementation
and their use in ROMS are presented in the Supplement. Examples of model behavior are presented in Section 3, and a
realistic application in the York River Estuary is presented in Section 4. Discussion and Conclusions are in Sections 5 and 6.
**2 Model Processes**
Flocculation is represented as a local process of aggregation and disaggregation that moves mass among the floc classes
within each model grid cell during a ROMS baroclinic time step. ROMS uses a split time step scheme that integrates over
several (ca. 20) depth-averaged (barotropic) time steps before the depth-dependent baroclinic equations are integrated
(Shchepetkin and McWilliams, 2005). Subsequent advection and mixing of floc particles is performed along with other
tracers (heat, salt, sand, biogeochemical constituents). The water column is coupled with the sediment bed via depositional
fluxes determined by near-bed concentrations, settling velocities, and threshold shear stresses; and via erosional fluxes
determined by bottom shear stresses, bulk and particle critical shear stresses for erosion, and sediment availability in the top,
active layer (Warner et al., 2008). The distribution of mass among the cohesive classes can change in the bed as flocs are
converted to denser aggregates. Deposition and erosion affect the mass of sediment classes in the stratigraphic record, which
can also be changed by biodiffusive mixing and a heuristic model of erodibility as a function of time and sediment depth.
Each of these processes is described below.
**2.1 Properties of Sediment, Seafloor, and Seabed**
ROMS accounts for two distinct types of sediment: non-cohesive sediment (e.g., sand) and cohesive sediment (e.g., mud).
The general framework used to represent sediment and the seabed is unchanged from Warner et al. (2008), except that the
expanded model requires additional variables to allow for both cohesive and non-cohesive classes. The number of sediment
classes is presently limited to twenty-two of each type by the input/output formats, but is otherwise only constrained by
computational resources. Each class must be classified as either non-cohesive or cohesive, and at least one class of one type
is required for sediment-transport modeling. Each class is associated with properties (diameter, density, critical shear stresses
for erosion and deposition, settling velocity) that are specified as input and remain constant throughout the model
calculations. Seafloor properties that describe the condition of the sediment surface are stored with spatial dimensions that
correspond to the horizontal model domain. Seafloor properties include representative values (geometric means) of sediment
properties in the top layer, including grain size, critical shear stress for erosion, settling velocity, and density; and properties
of the sediment surface, such as ripple height, ripple wavelength, and bottom roughness. Seabed properties (i.e. stratigraphy)
are tracked at each horizontal location and in each layer in the bed. The number of layers used to represent seabed properties
is specified as input and remains constant throughout the model run. The mass of each sediment class, bulk porosity, and
average sediment age is stored for each bed layer. The layer thickness, which is calculated from porosity and the mass and
sediment density for each class is stored for convenience, as is the depth to the bottom of each layer. Additional information
for bulk critical shear stress is stored if the cohesive sediment formulation is being used.

## 2.2 Floc Model

Maerz et al. (2011) note that there are two approaches for representing particle sizes in models. Distribution-based models
use one value (e.g., the average or median) to represent the particle size distribution and sometimes floc density.
Distribution-based models are the most common: examples include Winterwerp (2006), Manning and Dyer (2007), and
Khelifa and Hill (2006). Van Leussen (1998) and Soulsby et al. (2013) provide reviews. In a numerical model, distribution-
based models require advection schemes that allow for spatial and temporal variation of settling velocity. In contrast, size-
class-based models represent the particle population by apportioning mass among a discrete number of size classes through
semi-empirical descriptions of break-up and aggregation, following the pioneering work of Smoluchowski (1917). Recent
examples include Hill and Nowell (1995), Xu et al. (2008), and Verney et al. (2011). One advantage of class-based models it
that simpler and more efficient advection schemes designed for constant settling velocities can be used for each class in turn.
The tradeoff is that (many) more size classes are required. Our implementation takes the second approach, and we
characterize sediment and floc distributions with several (7 – 20+) classes, each with fixed characteristics including size, floc
density, and settling velocity. This allows us to take advantage of the efficient settling flux algorithms in ROMS.

### 2.2.1. Water-Column Processes

We implemented the floc model FLOCMOD (Verney et al., 2011) in ROMS to model changes in settling velocity and
particle size caused by aggregation and disaggregation. The floc model is a zero-dimensional model that is locally integrated
over the baroclinic time step, from initial to final conditions, in every cell of the ROMS model. After the floc populations are
updated, the normal settling, advection, and diffusion routines used for tracers (heat, salt, flocs or other sediment,
biogeochemical constituents) in ROMS are advanced, with flux boundary conditions at the bed (erosion or deposition) and
zero-flux conditions at the surface. FLOCMOD is a population model (Smoluchowski, 1917) based on a finite number of
size classes with representative floc diameters $D_f$ (m). The model requires a relationship between floc size and floc density
$\rho_f$ (kg/m$^3$) that is related to the primary disaggregated particle diameter $D_p$ (m) and density $\rho_s$ (kg/m$^3$) through a fractal
dimension $n_f$ (dimensionless; Kranenburg, 1994) according to
$$\rho_f = \rho_w + \left(\rho_s - \rho_w\right)\left(\frac{D_f}{D_p}\right)^{n_f - 3} \tag{1}$$

where $\rho_w$ (kg/m$^3$) is the density of the interstitial water in the flocs. The fractal dimension for natural flocs is typically close to 2.1 (Tambo and Watanabe, 1979; Kranenburg, 1994). Floc densities increase as $n_f$ increases, and at $n_f = 3$, the flocs are solid particles with $\rho_f = \rho_s$. All cohesive sediment classes are treated as flocs when the floc model is invoked, and the processes of aggregation and disaggregation can shift mass of suspended sediment from one class to another. The floc model is formulated as a Lagrangian process that takes place within a model cell over a baroclinic model time step while conserving suspended mass in that cell, similar to the way that reaction terms are included in biogeochemical models (for example, Fennel et al., 2006). FLOCMOD simulates aggregation from two-particle collisions caused by either shear or differential settling, and disaggregation caused by turbulence shear and/or collisions. The rate of change in the number concentration $N(k)$ (m$^{-3}$) of particles in the $k^{th}$ floc class is controlled by a coupled set of $k$ of differential equations

$$\frac{dN(k)}{dt} = G_a(k) + G_{bs}(k) + G_{bc}(k) - L_a(k) - L_{bs}(k) - L_{bc}(k) \qquad (2)$$

where $G$ and $L$ terms (m$^{-3}$s$^{-1}$) represent gain and loss of mass by the three processes denoted by subscripts: $a$ (aggregation), $bs$ (breakup caused by shear), and $bc$ (breakup caused by collisions). Equations 2 are integrated explicitly using adjustable time steps that may be as long as the baroclinic model time step, but are decreased automatically when necessary to ensure stability and maintain positive particle number concentrations. Particle number concentrations $N(k)$ are related to suspended mass concentrations $C_m(k)$ (kg/m$^3$) via the volume and density of individual flocs. The aggregation and disaggregation terms (Verney et al., 2011) both depend on local rates of turbulence shear, which are calculated from the turbulence submodel in ROMS. Details of these processes are described in the Supplement.

The floc model introduces several parameters (see Supplement), some of which have been evaluated by Verney et al. (2011). These parameters are specified by the user. The equilibrium floc size depends on the ratio of aggregation to breakup parameters, and the rate of floc formation and destruction depends on their magnitudes (Winterwerp, 1999; 2002). The diameter, settling velocity, density, critical stress for erosion, and critical stress for deposition (described below) are required inputs for each sediment class, both cohesive and non-cohesive (see Supplement). The present implementation requires a fractal relationship between floc diameter and floc density (Kranenburg, 1994), and we have assumed a Stokes settling velocity. Alternative relationships between diameter and settling velocity, such as modified Stokes formula (e.g., Winterwerp, 2002; Winterwerp et al., 2002; Winterwerp et al., 2007; Droppo et al., 2005; Khelifa and Hill, 2006), could be used by adjusting input parameters, but alternative relationships between diameter and floc density (Khelifa and Hill, 2006; Nguyen and Chua, 2011) would require changes to the aggregation and disaggregation terms in FLOCMOD.

### 2.2.3. Changes in floc size distribution within the bed

Changes in the size-class distribution of flocs are expected once they have been incorporated into the seabed, in contrast to non-cohesive particles that retain their properties during cycles of erosion and deposition. For example, it seems unlikely that large, low-density flocs can be buried and later resuspended intact, and limited published observations suggest that material deposited as flocs can be eroded as denser, more angular aggregates (Stone et al., 2008). However, we find little guidance for constraining this process. We therefore have implemented floc evolution in the bed, a simple process that stipulates an equilibrium cohesive size-class distribution and an associated relaxation time scale. The time-varying size-class distribution in the bed tends toward the user-specified equilibrium distribution while conserving mass (see Supplement). If the equilibrium distribution includes more smaller, denser particles and less larger, less-dense particles than the depositing flocs, the particle population in the bed will evolve toward smaller, denser particles, changing the amount of material in the classes that are available for resuspension when a cohesive bed is eroded. Example cases presented below demonstrate the effect of this process and the associated time scale on floc distributions both in the bed and in the water column.

### 2.3. Bed – Water-Column Exchange

### 2.2.1. Fluxes into the bed – Critical shear stress for deposition

The settling flux of flocs (and all other size classes) into the bed (deposition) over a time step is calculated as $w_{s,k} \rho_k C_{v,k} \Delta t$ (kg m$^{-2}$, where $w_{s,k}$ (m/s), $\rho_k$ (kg/m$^3$), and $C_{v,k}$ (m$^3$/m$^3$) are settling velocities, floc (or particle) densities, and volume concentrations for the $k$th size class in the bottom-most water-column layer, respectively, and $\Delta t$ (s) is the baroclinic time step. An optional critical shear stress for deposition ($\tau_d$; Pa; Krone, 1962; Whitehouse et al., 2000; Spearman and Manning, 2008; Mehta, 2014) has been implemented for cohesive sediment. Deposition in our model is zero when the bottom stress $\tau_b$ (Pa) is greater than $\tau_d$. When $\tau_b$ is less than $\tau_d$, deposition increases linearly as $\tau_b$ decreases toward zero, behavior we call linear depositional flux (Whitehouse et al., 2000; see Supplement). A simpler alternative is to assume a full settling flux when $\tau_b < \tau_d$, which we call constant depositional flux, and which we have implemented as an option. According to Whitehouse et al. (2000), $\tau_d$ is typically about half the magnitude of the critical shear stress for erosion $\tau_c$, but is unrelated to that value. Mehta (2014, Equation 9.83) suggested a relationship between $\tau_d$ for larger particles, using $\tau_d$ values for the smallest particles in suspension and the ratio of diameters raised to an exponent that depends on sediment properties (see Supplement), citing Letter (2009) and Letter and Mehta (2011). The effect of a critical shear stress for deposition is to keep sediment in suspension in the bottom layer. This results in more material transported as suspended sediment and, for flocs, allows aggregation and disaggregation processes to continue.

**2.2.2. Fluxes out of the bed – Resuspension**

Resuspension is modelled as an erosional mass flux $E_{s,i}$ from the top (active) bed layer to the bottom-most water column cell (Ariathurai and Arulanandan, 1978; Warner et al., 2008) where

$$E_{s,i} = E_{0,i}(1-\phi)\frac{\tau_{sf} - \tau_{ce,i}}{\tau_{ce,i}}, \quad \text{when } \tau_{sf} > \tau_{ce,i} \tag{3}$$

where $E_0$ is a bed erodibility constant ($\text{kg m}^{-2}\,\text{s}^{-1}$), $\phi$ is porosity of the top bed layer, $\tau_{sf}$ is the skin-friction component of the bottom shear stress (Pa), $\tau_{ce}$ is the effective critical shear stress (Pa), and $i$ is an index for each sediment class. The total mass eroded over a time step is limited by amount of that sediment class in the top layer of the bed. The skin-friction component of the bottom shear stress is calculated using a wave-current bottom boundary layer model (Warner, 2008). The effective critical shear stress for non-cohesive sediment depends on grain characteristics, but $\tau_{ce}$ for cohesive beds is a bulk property of the bed, as discussed below in Section 2.5. The effective critical shear stress for mixed beds (i.e., non-cohesive grains in a cohesive matrix) varies, as described below in Section 2.6.

**2.4 Stratigraphy**

Stratigraphy serves two functions in the model as conditions change and sediment is added or removed from the bed: (1) to represent the mixture of sediment available at the sediment-water interface for use in bedload transport, sediment resuspension, and roughness calculations; and (2) to record the depositional history of sediment. Bookkeeping methods for tracking and recording stratigraphy must conserve sediment mass and must accurately record and preserve age, porosity, and other bulk properties that apply to each layer. Ideally, a layer could be produced for each time step in which deposition occurs, and a layer could be removed when cumulative erosion exceeds layer thickness. In practice, the design of many models is subject to computational constraints that limit resolution to a finite and relatively small number of layers. In ROMS, this number is declared at the beginning of the model run and cannot change. Thus, when deposition requires a new layer, or when erosion removes a layer, other layers must be split or merged so that the total number of layers remains unchanged. Where and when this is done determines the fidelity and utility of the modeled stratigraphic record. Some models have used a constant layer thickness (Harris and Wiberg, 2001); others (for example, ECOMSED) define layers as isochrons deposited within a fixed time interval (HydroQual, Inc., 2004). Our approach is most similar to that described by Le Hir et al. (2011) in that we allow mixing of deposited material into the top layer, and require a minimum thickness of newly formed layers, merging the bottom layers when a new layer is formed. Likewise, the bottom layer is split when erosion or thickening of the active layer, discussed below, reduces the number of layers. The sequence of layer calculations is described in detail in the Supplement.

A key component of the bed model is the active layer (Hirano, 1971), which is the thin (usually mm-scale), top-most layer of
the seabed that participates in exchanges of sediment with the overlying water. During each model time step, deposition and
erosion may contribute or remove mass from the active layer. Any stratigraphy in the active layer is lost by instantaneous
mixing (Merkel and Klopmann, 2012), but this is consistent with the original concept of Hirano (1971) and the need to
represent the spatially averaged surface sediment properties in a grid cell that represents a heterogeneous seabed. The
thickness of the active layer in ROMS scales with excess shear stress (Harris and Wiberg, 1997; Warner et al., 2008) and is
at least a few median grain diameters thick (Harris and Wiberg, 1997; see Supplement).

## 2.5 Bulk Critical Shear Stress for Erosion for Cohesive Sediment

An important difference between cohesive and non-cohesive sediment behavior is that the erodibility of cohesive sediment is
treated primarily as a bulk property of the bed, whereas the erodibility of non-cohesive sediment is treated as the property of
individual sediment classes. The erodibility of cohesive sediment often decreases with depth in the bed, resulting in depth-
limited erosion (Type 1 behavior according to Sanford and Maa, 2001). When the cohesive bed module is used, the
erodibility of cohesive beds depends on the bulk critical shear stress for erosion $\tau_{cb}$ (Pa), which is a property of the bed
layer, not individual sediment classes, and generally increases with depth in the bed. It also changes with time through
swelling and consolidation and, in the uppermost layer, is affected by erosion and deposition. The cohesive bed model tracks
these changes by updating profiles of $\tau_{cb}$ at each grid point during each baroclinic timestep.
There is no generally accepted physically based model for determining $\tau_{cb}$ from bed properties such as particle size,
mineralogy, and porosity. We adopted Sanford's (2008) heuristic approach based on the concept that the bulk critical shear
stress profile tends toward an equilibrium profile that depends on depth in the seabed (Figure 1) and must be determined a
priori. Erosion-chamber measurements (Sanford, 2008; Rinehimer et al., 2008; Dickhudt et al., 2009; Dickhudt et al., 2011;
Butman et al., 2014) have been used to define equilibrium bulk critical shear stress profiles $\tau_{cbeq}$ in terms of an exponential
profile defined by a slope and offset.
$$\tau_{cb\,eq} = a \exp\left[\left(\ln\left(z_\rho\right) - offset\right) / slope\right]$$
(4)

where $z_\rho$ (kg/m$^2$) is mass depth, the cumulative dry mass of sediment overlying a given depth in the bed. In Equation 3, *offset*
and *slope* have units of ln(kg/m$^2$), and $a = 1$ Pa kg$^{-1}$ m$^2$ is a dummy coefficient that produces the correct units of critical shear
stress. The mass depth at the bottom of each model layer $k$ is calculated as
$$z_\rho(k) = \sum_k \sum_i f_{i,k} \rho_i \Delta z_k$$
(5)

where the summations are computed over the $k$ bed layers and $i$ sediment classes, $f_i$ (dimensionless) is the fractional amount of sediment class $i$, $\rho_i$ (kg/m$^3$) is particle density in class $i$, and $\Delta z_k$ (m) is the thickness of layer $k$. Equation 3 can be written in terms of the power-law fits to erosion-chamber measurements presented by Dickhudt (2008) and Rinehimer et al. (2008; see Supplement). The instantaneous bulk critical shear stress profile is nudged over time scale $T_c$ or $T_s$ (s) toward the equilibrium profile to represent the effects of consolidation or swelling following perturbations caused by erosion or deposition. $T_c$ is the time scale for consolidation and is applied when the instantaneous profile is more erodible than the equilibrium value, while $T_s$ is the time scale for swelling and is applied when the instantaneous profile is less erodible than the equilibrium value. The consolidation time scale is usually chosen to be much shorter than the one associated with swelling (Sanford, 2008). New sediment deposited to the surface layer is assigned a bulk critical shear stress that may either be (1) held constant at a low value (Rinehimer et al. 2008), or (2) set at the instantaneous bed shear stress of the flow.

## 2.6 Mixed Sediment

Mixed-sediment processes occur when both cohesive and non-cohesive sediment are present, and are typically sensitive to the proportion of mud. Beds with very low mud content (<3%; Mitchener and Torfs, 1996) behave as non-cohesive sediment: erodibility is determined by particle critical shear stress, which is an intrinsic characteristic of each particle class. Non-cohesive beds may be winnowed and armored by selective erosion of the finer fraction. In contrast, beds with more than 3% to 15-30% (Mitchener and Torfs, 1996; Panagiotopoulos et al., 1997, van Ledden et al., 2004; Jacobs et al, 2011) mud content behave according to bulk properties that, in reality, depend on porosity, mineralogy, organic content, age, burial depth, etc., but that, in the model, are characterized by the bulk critical shear stress for erosion. Our approach to resuspension of mixed sediment is similar to that suggested by Le Hir et al. (2011) and Mengual et al. (2017). Mixed beds in the model have low to moderate mud content (3% to 30%, subject to user specification) and their critical shear stress in the model is a weighted combination of cohesive and non-cohesive values determined by the cohesive-behavior parameter $P_c$, which ranges from 0 (non-cohesive) to 1 (cohesive; see Supplement). Where $P_c = 0$, there is no cohesive behavior, and the particle shear stress $\tau_c$ for each sediment class is the effective critical shear stress $\tau_{ce}$ for that class. Where $P_c = 1$, the cohesive sediment algorithm is used, and the effective critical shear stress for each class is the greater of $\tau_c$ and the bulk critical shear stress $\tau_{cb}$. Between those limits, the effective critical shear stress for each sediment class is

$$\tau_{ce} = \max\left[ P_c \tau_{cb} + (1 - P_c)\tau_c, \quad \tau_c \right] \tag{6}$$

This is approach allows fine material (e.g., clay) to be easily resuspended when $P_c$ is low and only a small fraction of mud is present in an otherwise sandy bed, and it limits the flux to the amount available in the active mixed layer. It also allows non-cohesive silt or fine sand embedded in an otherwise muddy bed to be resuspended during bulk erosion events when $P_c$ is

high, and it provides a simple and smooth transition between these behaviors. The thickness of the active mixed layer is calculated as the thicker of the cohesive and non-cohesive estimates. Figure 2 illustrates mixed-bed behavior as the mud (in this case, clay-sized) fraction $f_c$ increases for a constant bottom stress of 0.12 Pa. At low $f_c$, $P_c$ is zero (Figure 2a), and clay and silt are easily eroded (high relative flux rates out of the bed; Figure 2c) because the particle critical shear stress for non-cohesive behavior of these fine particles is low (Figure 2b). The relative flux rates in Figure 2b are normalized by the fractional amount of each class and the erosion-rate coefficient; the actual erosional fluxes for clay content would be low at $P_c = 0$ because of the low clay content in the bed. As $f_c$ increases and the bed becomes more cohesive, relative erosion flux rates decline. When $f_c$ exceeds a critical value (0.2 in the example shown in Figure 2), the bed is completely cohesive and erosion fluxes are determined by bulk critical shear stress for erosion of cohesive sediment $\tau_{cb}$.

Non-cohesive sediment classes are subject to bedload transport when the bottom stress exceeds both the bulk critical shear stress of the top (active) layer and the particle critical shear stress for that class. In these cases, the transport-rate equations still calculate bedload transport based on excess shear stress associated with the non-cohesive particle critical shear stress, as described in Warner et al (2008). Cohesive classes are not subject to bedload transport; if the bulk critical shear stress of the bed is exceeded, we assume they will go directly into suspension.

**2.7 Bed Mixing**

Mixing of bed properties in sediment can be caused by benthic fauna (ingestion, defecation, or motion such as burrowing) or circulation of porewater, and tends to smooth gradients in stratigraphy and move material vertically in sediment. The model (e.g., Boudreau, 1997) assumes that mixing is a one-dimensional vertical diffusive process and neglects non-local and lateral mixing processes:

$$\frac{\partial C_v}{\partial t} = \frac{\partial}{\partial z}\left( D_b \frac{\partial C_v}{\partial z} \right) \tag{7}$$

where $C_v$ is the volume concentration of a conservative property (e.g., fractional concentration of sediment classes or porosity), $D_b$ is a (bio)diffusion coefficient ($m^2$/s) that may vary with depth in the bed (see below), and $z$ (m) is depth in the bed (zero at the sediment-water interface, positive downward). We have discretized Equation (7) using the varying bed thicknesses and solve it at each baroclinic time step using an implicit method that is stable and accurate (See Supplement).

Biodiffusivity is generally expected to decrease with depth in the sediment (Swift et al., 1994; 1996), but is often assumed to be uniform near the sediment-water interface. The typical depth of uniform mixing, based on worldwide estimates using radionuclide profiles from cores, is 9.8±4.5 cm (Boudreau, 1994). Rates of biodiffusion estimated from profiles of excess

$^{234}$Th on a muddy mid-shelf deposit off Palos Verdes (California, USA) varied from ~2 cm$^2$/yr to ~80 cm$^2$/yr (Wheatcroft
and Martin, 1996; Sherwood et al., 2002) and values from the literature range from 0.01 – 100 cm$^2$/yr (Boudreau, 1997;
Lecroart et al., 2010). The depth-dependent biodiffusion rate profile in the model must be specified for each horizontal grid
cell using a generalized shape described in the Supplement.
Representation of seabed properties, i.e. the stratigraphy, has been modified slightly from the framework presented in
Warner et al. (2008). The revised bed model gives the user latitude to control the resolution of the bed model through the
choice of new layer thickness and the number of bed layers, and avoids the mixing described by Merkel and Klopmann
(2012). The bookkeeping for bed layers is detailed in the Supplement. The main differences from previous versions of the
model (Warner et al., 2008) are the treatments of the second layer (immediately below the active layer) and the bottom layer.
During deposition, the new algorithm prevents the second layer from becoming thicker than a user-specifed value, which
results in thinner layers that can record changes in sediment composition inherited from the active layer as materials settle.
During erosion, the new algorithm splits off only a small portion of the bottom layer to create a new layer. This limits the
influence of the initial stratigraphy specified for the bottom layer and confines blurring of the stratigraphic record to the
bottommost layers. Our tests indicate the new approach provides a more informative record of stratigraphic changes.
Moriarty et al. (2017) used a similar approach to bed stratigraphy to preserve spatial gradients in sediment biogeochemistry.
**3 Demonstration Cases**
The following cases demonstrate the cohesive-sediment processes included in ROMS, explore model sensitivity to
parameters, and provide candidates for inter-model comparisons.
**3.1 Floc Model**
Tests using a quasi one-dimensional vertical implementation of ROMS were conducted to verify that the floc model was
implemented correctly and to gain some insight into model behavior under typical coastal conditions.
**3.1.1 Comparison with laboratory experiments**
Verney et al. (2011) compared results from FLOCMOD with a laboratory experiment of tidal-cycle variation in shear rate $G$.
We performed the same simulations in ROMS by initializing with the same floc model parameters. The model was run with
15 cohesive classes (instead of the 100 classes in the reference FLOCMOD experiment). Settling velocities were set to zero,
and the turbulent shear parameter $G(t)$ was specified, ranging from $G$=0 s$^{-1}$ at slack tide to $G$=12 s$^{-1}$ at peak flow. Periodic
lateral boundary conditions were used, effectively creating a zero-dimensional simulation where the only active process was

floc response to the changing turbulent shear. The class sizes were log-spaced between 4 and 1500 µm with floc densities derived from Equation 1 using $n_f = 1.9$. The suspended-sediment concentration was constant at 0.093 kg/m³, and it was initially all in the 120-µm class. Our results (Figure 3a) matched the cycles of floc diameter variation caused by aggregation (low $G$) and breakup (high $G$) shown in Figure 7 of Verney et al. (2011), with a 24-µm root-mean square (rms) difference from observations in mass-weighted mean diameter. As in the Verney et al. (2011) simulation, our model did not reproduce the dip in mean grain diameter at ~400 min, which may have been caused by settling of the larger flocs in the laboratory experiment.

We also compared our ROMS FLOCMOD implementation with laboratory experiments of the growth and breakup of flocs performed by Keyvani and Strom (2014) who used a constant sediment concentration of 0.05 kg/m³ and applied cycles of $G=15$ s⁻¹ that caused floc growth followed by long periods (15 h) of very strong turbulent shear rates ($G=400$ s⁻¹) that caused disaggregation. We simulated the first cycle of floc formation using the size classes, fractal dimension, and concentrations provided by Keyvani and Strom (2014), but varying the aggregation parameter $\alpha$ and the breakup parameter $\beta$ that determine the final equilibrium diameter. Our model results with $\alpha=0.1$ and $\beta=0.0135$ (Figure 3b) reproduced the observations with higher skill than the simple model used in their study. The same final diameter was obtained with $\alpha=0.45$ and $\beta=0.06$, but the equilibrium was attained more quickly than observed.

These comparisons with laboratory results indicated that our implementation of FLOCMOD in ROMS was correct and demonstrated that the model has useful skill in representing floc dynamics.

### 3.1.2. Comparison to equilibrium floc size

Simulations were conducted to further evaluate the ROMS implementation of FLOCMOD by comparing modeled equilibrium floc sizes to equilibrium floc sizes predicted by Winterwerp (2006). He argued that, in steady conditions, equilibrium floc sizes are determined by the fractal dimension $n_f$, ratio of aggregation rates and breakup rates, concentration $C$ (kg/m³), and turbulence shear rate $G$ (s⁻¹). The equilibrium median floc size $D_{50}$ (m) is given by

$$D_{50} = D_p + \frac{k_A}{k_B} \frac{C}{\sqrt{G}} \tag{8}$$

where $k_A$ and $k_B$ are aggregation and breakup coefficients, respectively (Winterwerp, 1998). The units of $k_A$ and $k_B$ depend on fractal dimensions, but the ratio has units of m⁴kg⁻¹s⁻¹/². We compared our FLOCMOD results with this theoretical relationship by running cases with steady conditions, $n_f = 2$, for a range of concentrations ($C = 0.1$ to 10 kg/m³), a range of shear rates ($G = 0.025$ to 100 s⁻¹), and several combinations of aggregation and breakup parameters $\alpha$ and $\beta$. The results show

that equilibrium floc size increases with concentration and decreases with turbulence shear rate, as expected (Figure 3c).
Equilibrium diameter is strongly controlled by concentration, and turbulence is more effective at reducing average diameter
at lower concentrations. The slope of the relationship between the equilibrium diameter and $C/\sqrt{G}$ varies with the ratio of
aggregation to breakup. Winterwerp (1998) suggested a slope of about $4\times10^3$ $\text{m}^4\text{kg}^{-1}\text{s}^{-1/2}$. Figure 3c demonstrates that a range
of slopes can be obtained by varying the ratio $\alpha/\beta$. The model reproduced the linear response predicted by Winterwerp
(1998) except near the largest sizes, where our upper limit in floc class size (5000 µm) distorted the statistics. Although not
shown in Figure 3c, the floc populations evolved at different rates, depending on $\alpha$ and $\beta$, as indicated in Figure 3b.

**3.1.3. Evolution to steady state**

Steady, uniform flow is a conceptually simple model test that demonstrates the hydrodynamics linking vertical profiles of
flow, evolution of the turbulent boundary layer, and bottom drag. The addition of floc dynamics creates a complicated and
instructive test case. The model set-up was a fully three-dimensional implementation with advection, diffusion, and settling
of the dynamically changing floc population. The vertical grid included 40 cells, but the horizontal aspect of the grid was
small (5 cells…just enough to accommodate the templates of the finite-difference formulations) and included lateral periodic
boundary conditions, so that anything advected out of the domain re-entered on the upstream side. This simulation, forced by
a constant sea-surface slope, is similar to the steady flow test examined by Winterwerp (2002, section 4.8.1), and produces a
linear Reynolds-stress profile increasing from zero at the surface to $\tau_b = -\rho_w gh\, ds/dx$ at the seabed, where $\tau_b$ (Pa) is
bottom shear stress, $g$ (m/s$^2$) is gravitational acceleration, $h$ (m) is water depth, and $ds/dx$ (m/m) is sea-surface slope. The
flow develops a logarithmic velocity profile $u = (u_*/\kappa)\ln(z/z_o)$, where $u$ (m/s) is velocity in the $x$ direction,
$u_* = \sqrt{\tau_b/\rho_w}$ is shear velocity (m/s), $\kappa = 0.41$ (dimensionless) is von Kármán's constant, $z$ (m) is elevation above the bed,
and $z_0$ (m) is the bottom roughness length. The final flow velocity near the surface is about 0.6 m/s. When non-cohesive
sediment is added (and erosion and deposition are set to zero), the suspended sediment concentrations for each size class
evolve into Rouse-like profiles where, at each elevation, downward settling is balanced by upward diffusion. The addition of
floc dynamics complicates the situation, because aggregation creates larger flocs with higher settling velocities. The larger
flocs tend to settle into regions of higher shear and higher concentration, where the higher shear tends to break them into
smaller flocs but the higher concentrations enhance aggregation. The size distribution, settling velocity, concentration, shear,
and turbulent diffusion evolve to a steady state under a dynamic balance. The resulting profiles of concentration and mass-
weighted average size and settling velocity are sensitive to both floc model parameters and modeled physical conditions
(water depth, bottom stress, turbulence model, total sediment in suspension).
We demonstrate this process using 22 floc classes with logarithmically spaced diameters ranging from 4 to 5000 $\mu$m (Figure
4). The initial vertical concentration profile was uniform at 0.2 kg/m³, all in the 8-$\mu$m class. The model started from rest, and
the initial response was slow particle settling in the nearly inviscid flow: concentrations, floc sizes, and settling velocities all
decreased near the surface (Figures 4a, b, and c). As the flow accelerated in the first two hours, turbulence generated by
shear at the bottom began to mix upward in the water column, diffusing settled material higher and facilitating collisions and
aggregation among flocs. Between hours 3 and 4, settling was enhanced by these newly formed larger flocs, as is apparent in
increases in average diameter and settling velocities, and reduced concentrations near the surface. Equilibrium was nearly
established by about hour 5. At the end of the model run, the total concentration profile decreased exponentially with
elevation (Figure 4d and 4g), but average size and settling velocities both decreased markedly in the bottom meter (Figures
4e and 4f), reflecting shear disaggregation that lead to increases in smaller flocs near the bottom (Figure 4g).
The time scales to achieve equilibrium in this simulation are comparable to tidal time scales, suggesting equilibrium is
unlikely in the real world, where forcing is time dependent and bottom conditions are spatially variable. The final condition
is sensitive to flow forcing, initial concentrations, and floc parameters. For example, when concentrations are higher, or
when the disaggregation parameter is increased (making the flocs more fragile), bottom-generated shear causes
disaggregation higher into the water column, and mid-depth maxima in diameter and settling velocity evolve. This steady
flow simulation is useful as both a standard test case and a reminder of the complexity of floc processes, even when the
hydrodynamics are relatively simple.
**3.1.4. Settling fluxes**
Interaction with the bed influences the evolution of the floc population in the water column by providing sources or sinks in
various size classes. We have experimented with several sediment-flux conditions from the water column to the seabed,
including settling fluxes, zero fluxes, and fluxes modulated by threshold stresses for deposition. Settling fluxes calculated as
$w_k \rho_k C_k \Delta t$ summed over each class $k$, is the default method used for non-cohesive sediment. Zero-flux boundary conditions
essentially treat the bottom water-column cell as a fluff layer, allowing flocs to accumulate by settling or mix out by
diffusion. Floc dynamics continue to operate in this layer, so the size distributions change with concentration and stress.
Settling fluxes modulated by stress thresholds for deposition allow flocs to deposit only under relatively quiescent
conditions. The model framework provides a variety of choices described in the Supplement, each with implications that
must be assessed in the context of the problem at hand. As expected, the conditions that reduced settling into the bed resulted
in higher sediment concentrations in the bottommost water-column layer and allowed for floc breakup by the enhanced near-
bottom turbulence.

### 3.1.5. Model sensitivity

A wide range of model runs (not presented here) have provided us with a qualitative sense of model performance. Model results respond as expected to physical parameters, such as mean concentration and shear rate (discussed above), as well as primary particle size and fractal dimension. Model results are also sensitive to model configuration, including the number of size classes, the size of vertical grid spacing, and the time step used. Our experience so far confirms that of Verney et al. (2011): a truncated distribution of about seven size classes provides qualitatively useful results, but the choice of size range and size distribution may change the results. The sensitivity to vertical grid resolution is particularly important in the bottommost layer, which has the highest concentrations and highest shear rates. Finer grid spacing near the bottom results in layers with higher shear and higher sediment concentrations, which cause local changes in the equilibrium floc sizes. Model time steps in our floc model tests are short, ranging from 10 to (more typically) 1 s. The adaptive sub-steps for aggregation and disaggregation were limited to a minimum of 0.5 s. At high concentrations ($> 0.2$ kg/m$^3$) and high shear rates, the results sometimes showed numerical instability, probably related to the explicit solution of Equations 2. Replacement of the solver for these equations with a faster and more robust method in the future should improve model stability.

### 3.2 Resuspension

Three cases are presented here to demonstrate the evolution of stratigraphy caused by resuspension and subsequent settling of sediment during time-dependent bottom shear stress events. They contrast model calculations using the non-cohesive and mixed-bed routines, and highlight the role of biodiffusion. These were one-dimensional (vertical) cases represented with small (~5 x 6 horizontal x 20 vertical cells), three-dimensional domains with flat bottoms and periodic lateral boundary conditions on all sides. They were forced with time-varying surface wind stress that generated time-dependent horizontal velocities and bottom stress, initialized with zero velocity and zero suspended-sediment concentration, and did not include floc dynamics in the water column.

### 3.2.1 Non-cohesive bed simulation

A non-cohesive bed simulation with a water depth of 20 m and periodic boundary conditions was used to demonstrate the generation and preservation of sand and silt stratigraphy during a resuspension and settling event (Figure 5). The model was forced with two stress events ~ 1.5 d apart and lasting 1.5 d and 1 d respectively. Four sediment classes, representing particles with nominal diameters of 4, 30, 62.5, and 140 $\mu$m, particle critical shear stresses of 0.05, 0.05, 0.1, and 0.1 Pa, and settling velocities of 0.1, 0.6, 2, and 8 mm s$^{-1}$ were used. Although the diameters of the first two sediment classes corresponded to mud, all sediment classes in this experiment were treated as non-cohesive material. The initial sediment bed

contained 41 layers, each 1 mm thick, and each holding equal fractions (25%) of the four sediment classes. New sediment
layers were constrained to be no more than 1 mm thick.
The first, larger stress event (maximum    = 1 Pa; Figure 5b), eroded 1.2 cm of bed, and expanded the active layer to a
thickness of 0.8 cm, so the bed was disturbed to a depth of 2 cm. Expansion of the active layer homogenized enough layers
to provide 0.8 cm of sediment, making more fine sediment available for resuspension. The finer fractions dominated the
suspended sediment in the water column, which contained only a small fraction of the coarsest sand (Figure 5a). When the
stress subsided, coarser sediment deposited first, while finer material remained suspended, producing thin layers of graded
bedding above the 2-cm limit of initial disturbance (Figure 5d).
The second stress pulse eroded the bed down to 1 cm but only resuspended minimal amounts of the 140- $\mu$m sand.
Deposition resumed after the second pulse subsided and, at the end of the simulation, some mud remained in the water
column (Figure 5a), leaving the bed with net erosion of 5 mm (Figure 5d). The finest material (4 $\mu$m) remained mostly in
suspension after five days. The final thickness of the bottom five layers was smaller than their initial value (1 mm), because,
to maintain a constant number of bed layers, the deepest layer was split each time a surface layer was formed during
deposition. The two stress pulses affected sediment texture down to 2 cm. Above this level, almost all of the finest class was
winnowed, and remained mostly in suspension while the other classes settled to the bed, so that the upper bed layers
developed a fining-upward storm layer. The bottom portion of the storm layer (1 – 2 cm depth) was a lag layer comprised of
the two coarsest classes, both because these resisted erosion and because the sand that did erode settled to the bed quickly
when shear stress decreased.
**3.2.2 Mixed bed simulation**
This case examined the stratigraphic consequences of cohesive behavior resulting from a single bottom-stress event (Figure
6). The model configuration was similar to the previous example. The same sediment classes were used, but the two finest (4
and 30 $\mu$m) were treated as cohesive mud, while the other two remained non-cohesive (sand). The fraction of cohesive
sediment ($f_c$ = 0.5) exceeded the chosen non-cohesive threshold ($f_{nc}$ threshold = 0.2), so the bed behaved as if it were
completely cohesive. The cohesive formulation required the initialization of an equilibrium bulk critical stress profile for
erosion. We chose parameters within the range of sensitivities studied by Rinehimer et al. (2008) and specified an
equilibrium profile with a slope = 2 ln(kg/m$^2$) and an offset of 3.4 ln(kg/m$^2$), with a minimum value of 0.03 Pa and a
maximum of 1.5 Pa (dashed magenta line in Figure 6b) and initialized the model with this profile (solid purple line in Figure
6b). The time scale for consolidation was set to $T_c$= 8 hours. The swelling time scale was chosen to be 100 times longer than
consolidation ($T_s$ = 33 days). A time series of bed stress was imposed (Figure 6a), and the bed responded initially by
eroding. As the imposed stress waned starting at day 37, sediment settled to the bed causing deposition. The initial rapid
increase in bottom stress during the first 0.7 days (Figure 6a) exceeded the critical stress of the bed to a depth of 2.4 cm (red
line in Figure 6c), causing resuspension and erosion of the top 5 mm of the bed. In this case, the amount of material eroded
was limited by the erosion rate coefficient. The equilibrium critical stress profile, which has a static shape, shifted down with
the sediment-water interface (compare dashed magenta line in Figures 6b, c). After the initial erosion, the instantaneous
critical stress profile tended toward the equilibrium critical stress profile over the slow swelling time scale of 33 days,
rendering the bed progressively more erodible (compare Figures 6c, d). The process of swelling, while slow, rendered the
bed more erodible, and an additional 2-3 mm of sediment was removed by day 32. By day 38, the stress had waned and 4
mm of sediment had redeposited (Figure 6d). The equilibrium critical stress profile had shifted upward with the bed surface,
causing the instantaneous critical stress to increase over the short compaction time scale. The final instantaneous critical
shear stress profile (Figure 6e) had almost reached the long-term equilibrium everywhere except in the most recent deposits.
This case exemplifies the sequence of depth-limited erosion, deposition, and compaction that characterizes the response of
mixed and cohesive sediment in the model.
**3.2.3 Biodiffusion simulations**
We validated the numerical performance of the biodiffusion algorithms using two analytical test cases with a realistic range
of parameters. The implicit numerical solution is unconditionally stable and conserves mass to within $10^{-8}$ %, but the
accuracy depends on time step, gradients in biodiffusivity, and bed thickness. Typical RMS differences in the fractional
amount of sediment in a particular class between the numerical solutions and the analytical solutions ranged from $10^{-2}$ to $10^{-6}$.
We found that, for modeled beds 5 m thick, solutions improved as layer thickness decreased from 50 to 5 cm, but beyond
that, higher resolution did not substantially improve the solution. Even in the worst case, where the numerical solution was
off by 1%, it was much more precise than our estimates of biodiffusivity coefficients.
Four cases are presented to demonstrate bed mixing (Figure 7). The first two used the same configuration as in the non-
cohesive (Figures 5, 7a) and mixed-bed simulations (Figures 6d, 7b). The second two were identical to the mixed-bed case
except that biodiffusive mixing was enabled. The biodiffusivity profile used was similar to that proposed for the mid-shelf
deposit offshore of Palos Verdes, CA (Sherwood et al., 2002) that had a constant diffusivity $D_{bs}$ from the sediment-water
interface down to 2 mm, an exponential decrease between 2 mm and 8 mm, and a linear decrease to zero at 1 cm depth.
These two cases differed in their biodiffusion coefficients: a) the first used relatively large biodiffusion coefficients ($D_{bs}$ =
$10^{-5}$ m$^2$s$^{-1}$); b) the second used smaller values ($D_{bs} = 10^{-10}$ m$^2$s$^{-1}$).
The resulting stratigraphy after the five-day simulation (Figure 7) indicates that mixing in the case with large biodiffusivity
(Figure 7c) tended to smooth all gradients rapidly and only during depositional conditions was the vertical structure of grain
size fractions preserved. Some sediment remained in suspension in all four cases, which was reflected in the final bed
elevation. The resulting top 1 cm of the bed was always well mixed and the depth of the disturbed sediment at the end of the
simulation was deeper (2.5 cm) in this case than in the other simulations. Sediment deeper than 2.5 cm below the surface was
undisturbed: it was beyond the reach of erosion, active-layer formation, and biodiffusion. The biodiffusive mixing increased
recruitment of fine sediment into the surface active layer during erosion, resulting in increased concentrations in the water
column (not shown) compared to the mixed bed case without biodiffusion.
The case with a smaller biodiffusion coefficient (Figure 7d) developed stratigraphy intermediate to those cases with large
and zero biodiffusion. The depth of disturbed sediment was 2.3 cm and the transition between redeposited sand and mud was
smooth with coarse sand being present at the surface of the bed. This gradual size gradation was intermediate to the sharp
jump in the fractional distribution between mostly sandy layers and predominantly muddy layers produced in cases that
neglected mixing (Figure 7a,b) and the smooth gradient produced by the strong mixing case (Figure 7c).
**3.3 Estuarine Turbidity Maxima**
High concentrations of suspended sediment often occur near the salt front in estuaries, forming estuary turbidity maxima
(ETM). We present ETM test cases that simulated sediment transport in a two-dimensional (longitudinal and vertical) salt-
wedge estuary with tidal and riverine forcing. The cases investigated the formation of cohesive deposits beneath the ETM
with and without floc dynamics. The first case, without floc dynamics but with a mixed bed, is presented here. The second
case, presented below, adds floc dynamics. The model was forced with a 12-hour tidal oscillation modulated with a 14-day
spring-neap cycle. The idealized estuary was 100-km long with a sloping bottom 4 m deep at the head of the estuary and 10
m deep at the mouth (Figure 8a). In all cases, the simulations were run for twenty tidal cycles. Two non-cohesive sediment
classes (180- and 250-$\mu$m diameter) were represented with equal initial bed fractions (50% of each). One cohesive fraction
(37 $\mu$m, $\rho_f$ = 1200 kg/m$^3$, $w_s$ = 0.13 mm/s) was included, with an initial uniform suspended-sediment concentration of 1
kg/m$^3$. The bed was initialized without any cohesive sediment, so it initially behaved non-cohesively. Later in the simulation,
bed behavior became mixed as suspended mud settled and was incorporated into the initially sandy bed. The chosen
equilibrium bulk critical shear stress profile (Equation 3) had *slope* = 5 ln(kg/m2) and *offset* = 2 ln(kg/m$^2$), with a minimum
value of 0.05 Pa and a maximum of 2.2 Pa. The time scale for consolidation was set to $T_c$=8 hours (Sanford, 2008;
Rinehimer, 2008), and the swelling time scale was set to $T_s$=33 days.
During the simulations, salinity and suspended-sediment field evolved into dynamic equilibria that were repeated over
consecutive tides. An estuarine turbidity maximum (ETM) developed between 10 km and 60 km from the mouth of the
estuary (Figure 8a) in the salt wedge generated by gravitational circulation and tidal straining (Burchard and Baumert, 1998;
MacCready and Geyer, 2001). Elevated suspended-sediment concentrations ranging from 0.7 to 2.05 kg/m$^3$ occupied most of
the bottom layer and extended to mid-depth. No floc dynamics were included, so all of the suspended material depicted in
Figure 8a was in the 37-$\mu$m class.
The second case was identical, except that it included floc dynamics. Fifteen cohesive (floc) classes and the two non-
cohesive (sand) classes were included. Floc-class diameters were logarithmically spaced, ranging from 20 to 1500 $\mu$m, with
floc densities ranging from 1350 to 1029.3 kg/m$^3$, and settling velocities ranging from 0.078 to 5.31 mm/s, commensurate
with Equation 1 with fractal dimension $n_f = 2$. The suspended-sediment concentration field was initialized with a uniform
concentration of 1 kg/m$^3$, all in the 37-$\mu$m class. The resulting ETM (Figure 8b) extended farther up-estuary and contained
much lower concentrations (0.1 to 0.5 kg/m$^3$ in most of the salt wedge, with a thin layer of higher concentrations (2.1 kg/m$^3$)
in the bottom layer (bottom 5% of the water column). The second layer (5 – 10% of the water column) had concentrations
about half of the bottom layer. The bed sediment response for the two cases also differed. In the no-floc case, the ETM
deposit was slightly thinner, located closer to the mouth, and varied less from slack to flood (Figure 8c). Floc dynamics
created large tidal variations in the size of bed material (Figure 8d), which ranged up to 600 $\mu$m as flocs deposited during
slack, and decreased to 37 $\mu$m as flocs were resuspended during flood. The behavior in the unflocculated case was less
intuitive. Over the course of the simulation, enough fine material accumulated beneath the ETM to cause the bed to behave
cohesively, but the top, active layer remained mostly non-cohesive. During flood tide, bottom stresses were sufficient to
resuspend the non-cohesive 70 $\mu$m material, leaving the cohesive 37 $\mu$m material on the bed. Thus, in both cases, the bed
became finer during period of higher stress, but for different reasons. The two cases highlight the model-dependent changes
in location (driven primarily by settling velocities) and size distributions (driven by floc dynamics) of the ETM.
We next expanded the numerical experiment, using six floc cases to elucidate the effects of floc dynamics in the idealized
estuary (Table 1). The two-dimensional model domain was the same as the ETM case described above. Three types of floc
behavior in the seabed were investigated: (1) no changes in size distribution occurred in the bed; (2) the floc evolution
process in the bed was invoked, which nudged all cohesive sediment into the 20-$\mu$m class over a long time scale (50 hours);
and (3) the floc evolution process was invoked with a short time scale (5 hours). Additionally, three other combinations of
aggregation ( $\alpha$ ) and disaggregation ( $\beta$ ) rates were used with the slow floc evolution in the bed rate to explore floc
processes in the water column (Table 1). The following six metrics were compared at the location of the maximum depth-
mean suspended-sediment concentration (SSC): depth-mean SSC; maximum SSC; median size ($D_{50}$); 12-h mean of the $D_{50}$;
depth-mean settling velocity $w_s$; and depth-mean $w_s$ averaged over a 12-h tidal period (Table 1). The median size and mean
settling velocities were weighted by the mass in each class. Also listed in Table 1 are the locus of the maximum deposition,
the thickness at that location, and the median size of deposited material at that location.
Mean SSC in the ETM did not vary significantly among the floc cases, but the maximum SSC (located lower in the water
column) increased when the ratio of aggregation rate / disaggregation rate $\alpha$ / $\beta$ was higher, which led to larger, faster-
settling flocs. Among the four cases (3 – 6) with slow floc evolution rates in the bed, settling velocities, maximum SSC, and
floc size covaried. The locus of maximum deposition of ETM material was insensitive to the algorithms for floc evolution in
the bed (cases 1 – 3), and most sensitive to the overall floc rates. The range of ETM locations is listed in Table 6 to highlight
the cases where ETM location varied. The case with lowest floc rates (case 5) produced the farthest upriver deposit, with the
most variation in the location of the maximum. The case with the highest settling velocities (case 6) produced deposits
closest to the estuary mouth. Overall, the simulated ETM was more sensitive to changes in floc parameters than to prescribed
behavior of the floc evolution in the seabed, and the greatest effect of varying floc dynamics was the vertical location of the
ETM, which was controlled by floc size and settling velocity.
**4 Realistic Application: York River Estuary**
This section demonstrates the cohesive sediment bed model in a realistic domain representing the York River, a sub estuary
of Chesapeake Bay (Figure 9). Recent modeling efforts have focused on this location as part of a program aimed at exploring
links between cohesive sediment behavior, benthic ecology, and light attenuation. As part of this program, colleagues have
obtained complementary field observations there, which have been especially focused on the two locations off Gloucester
Point and Clay Bank, VA (e.g. Dickhudt et al. 2009, 2011; Cartwright et al. 2013). The implementation presented here is
similar to the three-dimensional model developed by Fall et al. (2014) that accounted for circulation, sediment transport, and
a cohesive bed. While this model neglects flocculation, information obtained by field observations such as Cartwright et al.
(2013) have been consulted for guidance in setting settling velocities of the cohesive particles. The model is run assuming
muddy behavior of the bed, and neglecting mixed bed processes, because the majority of sediment transport within the York
River channels consists of fine-grained material. We found that it was important to modify the sediment bed layering
management scheme, as discussed in section 5 below, to resolve the high gradients in bed erodibility evident in the sediment
bed model (i.e. Fall et all 2014) and data (i.e. Dickhudt et al. 2009, 2011).
In this implementation, sediment deposited to the bed provided an easily erodible layer with an assumed low critical stress, $\tau_c$
= 0.05 Pa. The modeled sediment bed erodibility and suspended-sediment concentrations both were found to be sensitive to
parameterization of the equilibrium critical stress profile, and to the consolidation and swelling timescales used (Fall et al.,
2014). Here we present a case similar to that shown by Fall et al. (2014), but that differs mainly in terms of the sediment bed
initialization. The equilibrium critical stress profile was chosen as $\tau_{cbeq} = z_p^{0.62}$ which was a power-law fit of erodibility
experiments performed by Dickhudt (2008) on field-collected cores in September 2007 (Rinehimer et al., 2008). Swelling
and consolidation timescales of 1 day and 50 days, respectively, were used. Both the porosity ( $\phi = 0.9$ ) and the erosion rate
parameter $E_0 = 0.03$ kg/(m$^2$ s Pa) were held constant. A zero-gradient condition was applied for suspended-sediment
concentration at the open boundary where the York River meets Chesapeake Bay. Six sediment classes that had settling
velocities ranging from 0.032 to 10 mm/s were used. To initialize the seabed, they were distributed in equal fractions
throughout the model domain in a 20-layer sediment bed that had a total thickness of 1 m, with all but the bottom layer being
thin (0.1 mm). In this way, the model was initialized with a sediment bed that had high vertical resolution (0.1 mm) in the
upper ~2 cm, underlain by a thick layer (~1 m) sediment. This created high vertical resolution in the bulk critical shear stress
profile near the sediment – water interface, while still providing a fairly large pool of sediment so that erosional locations
retained some sediment in the seabed throughout the model run.  Bed critical stress was initialized everywhere to be constant
(0.05 Pa) with depth, and quickly evolved to the equilibrium critical shear stress profile at the compaction time scale of a few
days. The model was run to represent two months using the sixty-year median freshwater flow of 67 m$^3$/s and a spring-neap
tidal cycle with 0.2-m neap amplitude and 0.4-m spring amplitude.
The initially uniform bed evolved during the 60-day model run, developing areas of high sediment erodibility along the
shoals of the estuary and channel flanks (Figure 10a). In general, sediment was removed from the main channel, which
developed reduced erodibility (Figure 10a). At the Gloucester Point site, the initial bed evolved to become less erodible, with
a critical shear stress at the seabed that exceeded the equilibrium values specified for the model (Figure 10a). Conversely, at
the Clay Bank field site, conditions were variable in space. Sediment deposited on the shoal area, which evolved to enhanced
erodibility (Figure 10a). Within the channel, however, the equilibrium critical stress for erosion was often exceeded,
resulting in a strongly eroded sediment bed having larger values of critical shear at the sediment surface (Figure 10a).
Resuspension and transport also changed the spatial distribution of sediment classes, with the erosional areas retaining only
the coarser, faster-settling classes, while depositional areas retained finer-grained, slower-settling particles (Figure 10 b, c).
These patterns, with coarse lag layers and reduced erodibility in the channels relative to the shoals, are consistent with the
known grain size distributions and properties of the York River Estuary.
**5 Discussion**
The model algorithms presented here were motivated by the need to improve the representations of sediment dynamics in
numerical models of fine-grained and mixed-sediment environments. The improvements were implemented in the COAWST

version of ROMS, which provides a framework for realistic two-way nested models with forcing from meteorology (WRF; Michalakes et al., 2001) and waves (either SWAN: Booj et al., 1999; or WaveWatch III; Tolman et al., 2014). Waves, in particular, play an important role in cohesive sediment dynamics through wave-enhanced bottom shear stresses, wave-induced near-bottom turbulence, and wave-induced nearshore circulation, but wave-induced fluid-mud layer processes are not represented. ROMS includes options for several turbulence sub-models (e.g., $k - \varepsilon$, $k - \omega$, Mellor-Yamada) and wave-current bottom-boundary layer sub-models that allow us to calculate fields of shear velocity $G$. Implementation of FLOCMOD in this framework provides a platform for numerical experiments and real-world applications of a full-featured floc model.

The primary role of the floc model is to simulate the dynamical response of particle settling velocities to spatial and temporal variations in shear and suspended-sediment concentrations. This can also be achieved with simpler and computationally more efficient parameterization in many applications. What are the advantages of the complex and much slower model implemented here? There are several. The floc model provides fields of particles with dynamically varying density and number of primary particles, which allow calculation of the acoustic and optical responses of the particle fields. In turn, this allows direct comparison with field measurements of light attenuation, optical backscatterance, and acoustic backscatterance, the de facto proxies for suspended-sediment concentration. This also allows calculation of derived properties in the water column, including light penetration and diver visibility. Finally, the modeled particle properties can be used in geochemical calculations that require estimates of particle radius, porosity, and reactive surface area. Depending on the application, this additional information may justify the computational expense of the floc model.

The cohesive bed model provides a heuristic but demonstrably useful tool for representing muddy and mixed beds. The cohesive bed framework captures the most important aspects of muddy environment: limitations on erosion caused by increased bed strength with depth in the sediment, and changes toward user-defined equilibrium conditions as deposited (or eroded) beds age. The physical processes of self-compaction and associated changes in porosity and bed strength are not modeled, but the framework of particle-class and bed-layer variables are designed to accommodate a compaction algorithm. The equilibrium profile method implemented here adds little computational expense, but allows the model to represent depth-limited erosion, a key property of many cohesive beds.

Modeling stratigraphy effectively is challenging. Although conserving sediment mass among a fixed number of layers is straightforward, it has proven difficult to devise a robust and efficient method that records relevant stratigraphic events in a modeled sediment bed over the wide range of conditions that occur in coastal domains. For both sediment transport and sediment bed geochemistry (i.e. Moriarty et al. 2017; Birchler et al. 2018), it can be important for the sediment bed model to achieve its highest vertical resolution near the sediment – water interface, but the original ROMS sediment bed model did not meet that goal when the sediment bed was subject to frequent or repeated cycles of erosion. The modifications we have

made to the bed-layer management have improved the fidelity with which we can record stratigraphic events in the model layers, particularly at the sediment – water interface. Inclusion of biodiffusive mixing is important for environments where biological activity is rapid, compared with sedimentation or physical reworking. Additionally, for problems of sediment geochemistry, it is important to account for mixing of both particulate matter and porewater. Expansion of the ROMS sediment bed model to include diffusive mixing facilitates its use for interdisciplinary problems (i.e. Moriarty et al. 2017; Birchler et al. 2018). The choice of appropriate mixing parameters remains a challenge, especially when considering the spatial and seasonal heterogeneity of biological activity.

Overall, the cohesive and mixed-bed algorithms we have introduced in ROMS provide tools that should be useful for both numerical experimentations and realistic applications for fine-grained, and mixed-bed environments. The model applies to dilute suspensions at high Reynolds number (fully turbulent flow) because the turbulence sub-models do not account for particle influences on turbulence dissipation or momentum transfer (e.g., Hsu et al., 2003; Le Hir et al., 2001; Mehta, 1991; 2014), so fluid muds and non-Newtonian flows are not represented. We have not quantified the sediment concentrations or range of hydrodynamic parameters for which the model approximations are valid, but a common boundary for fluid mud (where viscoplastic properties become important) is 10 kg/m$^3$ (Einstein and Krone, 1962; Kirby, 1988). Other processes associated with cohesive or mixed sediment that have not been included are: flow-induced infiltration of fine material into a porous bed (Huettel et al., 1999); changes to the erodibility of mud that has been exposed at low tide (e.g., Paterson et al, 1990; Pilditch et al., 2008) or changes to erodibility caused by flora or fauna (e.g., de Boer, 1981; de Deckere et al., 2001; Malarkey et al., 2015; Parsons et al, 2016). It is important to note that the mass settling fluxes of mixed (sand + mud) suspensions may be overestimated if their interactions are not considered, as is the case in the approach taken here (Manning et al., 2010, Manning et al., 2011; Spearman et al., 2011). Nonetheless, our implementation of flocculation, bed consolidation, and bed-mixing modules enhance the utility of the ROMS sediment model for interdisciplinary studies including ecosystem feedbacks (light attenuation, biogeochemistry), and contaminant transport.

## 6 Conclusion

This paper describes three ways in which the sediment model of Warner et al. (2008) has been enhanced, allowing simulations to be made for non-cohesive, cohesive, and mixed sediment and allowing it to be applied in a wider range of studies. A flocculation model has been added, following Verney et al. (2011). The cohesive bed model developed by Sanford (2008) has been added, allowing the erodibility of the sediment bed to evolve in response to the erosional and depositional history. Mixing between bed layers has been implemented as biodiffusion using a user-specified diffusion coefficient profile. In addition, the sediment bed layering routine has been modified so that bed layers maintain a high resolution near the sediment water interface, as demonstrated by both our idealized and realistic case studies presented here. The paper presents

results of model runs that test and demonstrate these new features and to show their application to real-world systems. The
authors encourage the coastal modeling community to use, evaluate, and improve upon the new routines.
**Code and Data Availability**
The algorithms described here have been implemented in ROMS (version 3.6) distributed with the Coupled Ocean
Atmosphere Waves Sediment-Transport Modeling System (COAWST, Subversion repository revision number XXXX).
COAWST is an open-source community modeling system with a Subversion source-control system maintained by John C.
Warner (jcwarner@usgs.gov) and distributed under the MIT/X License (Warner et al., 2010). The COAWST distribution
files contain source code derived from ROMS, WRF, SWAN, MCT, and SCRIP, along with Matlab code, examples, and a
User's Manual.
**Supplement Link (supplied by Copernicus)**
**Team List**
**Author Contribution**
C.R. Sherwood and A. Aretxabaleta shared development of the model code and test cases and most of the manuscript
preparation. J.P. Rineheimer was an early user of the cohesive bed model and, along with C.K. Harris, developed the York
River application. R. Verney graciously supplied his FORTRAN version of FLOCMOD and helped with adaptation for
ROMS. B. Ferré contributed to the early development and application of the model. All authors contributed to the final
version.
**Competing Interests**
The authors declare that they have no conflict of interest.
**Disclaimer**
Use of firm and product names is for descriptive purposes only and does not imply endorsement by the U.S. Government.
**Special Issue Statement - None**
**Acknlowledgements**
The authors thank Jeremy Spearman, Alexis Beudin, Julia Moriarity and five anonymous reviewers (two from *Ocean*
*Dynamics* and three from *Geoscientific Model Development*) for helpful comments on earlier drafts of this manuscript. This
work was been supported by the U.S. Geological Survey, Coastal and Marine Geology Program and the National Ocean
Partnership Program. C.K. Harris was supported by NSF (OCE-1459708, OCE-1061781, and OCE-0536572). This is
contribution number XXXX of the Virginia Institute of Marine Sciences. B. Ferré is affiliated with the Centre of Excellence:
Arctic Gas hydrate, Environment and Climate (CAGE) funded by the Norwegian Research Council (grant no. 223259). The
model code is implemented in ROMS version 3.6, as distributed with the COAWST modeling system (Subversion repository
revision 1179; Warner et al., 2010), and is freely available by request to John C. Warner (jcwarner@usgs.gov) at the U.S.
Geological Survey.

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

**Table**

**Table 1. Characteristics of the estuary turbidity maxima for seven cases under different flocculation conditions.**

| Case | 0 | 1 | 2 | 3 | 4 | 5 | 6 |
|---|---|---|---|---|---|---|---|
| | No flocs | $\alpha = 0.35$ $\beta = 0.15$ no floc evol. | $\alpha = 0.35$ $\beta = 0.15$ floc evol., 5 h | $\alpha = 0.35$ $\beta = 0.15$ floc evol., 50 h | $\alpha = 0.45$ $\beta = 0.10$ floc evol., 50 h | $\alpha = 0.25$ $\beta = 0.20$ floc evol., 50 h | $\alpha = 0.35$ $\beta = 0.34$ floc evol., 50 h |
| Mean SSC @ maximum $(kg/m^3)$ | 1.23 | 0.46 | 0.45 | 0.45 | 0.45 | 0.46 | 0.46 |
| Maximum SSC $(kg/m^3)$ | 3.1 | 3.6 | 3.7 | 3.7 | 4.1 | 3.2 | 2.9 |
| $D_{50}$ at SSC maximum $(\mu m)$ | 37 | 539 | 529 | 529 | 622 | 426 | 384 |
| $D_{50}$ at SSC maximum; 12-h mean $(\mu m)$ | 37 | 255 | 249 | 250 | 325 | 181 | 167 |
| $w_s$ at SSC maximum (mm/s) | 0.13 | 1.91 | 1.87 | 1.87 | 2.2 | 1.51 | 1.36 |
| $w_s$ at SSC maximum; 12-h mean (mm/s) | 0.13 | 0.90 | 0.88 | 0.89 | 1.15 | 0.64 | 0.59 |
| Locus of maximum deposition (km from ocean boundary) | $80 \pm 30$ | $19 \pm 11$ | $18 \pm 10$ | $18 \pm 11$ | $19 \pm 10$ | $79 \pm 69$ | $16 \pm 6$ |
| Maximum deposit thickness (mm) | $4.2 \pm 5.8$ | $31.6 \pm 12.8$ | $25.8 \pm 10.1$ | $26.1 \pm 10.4$ | $27.1 \pm 10.9$ | $5 \pm 10.1$ | $25 \pm 10.2$ |
| Maximum deposit $D_{50}$ $(\mu m)$ | $18.5 \pm 0$ | $218 \pm 87.1$ | $40.9 \pm 71.3$ | $75.5 \pm 76.1$ | $92.9 \pm 94.2$ | $69.5 \pm 89.9$ | $25.4 \pm 40.4$ |


**Figures**

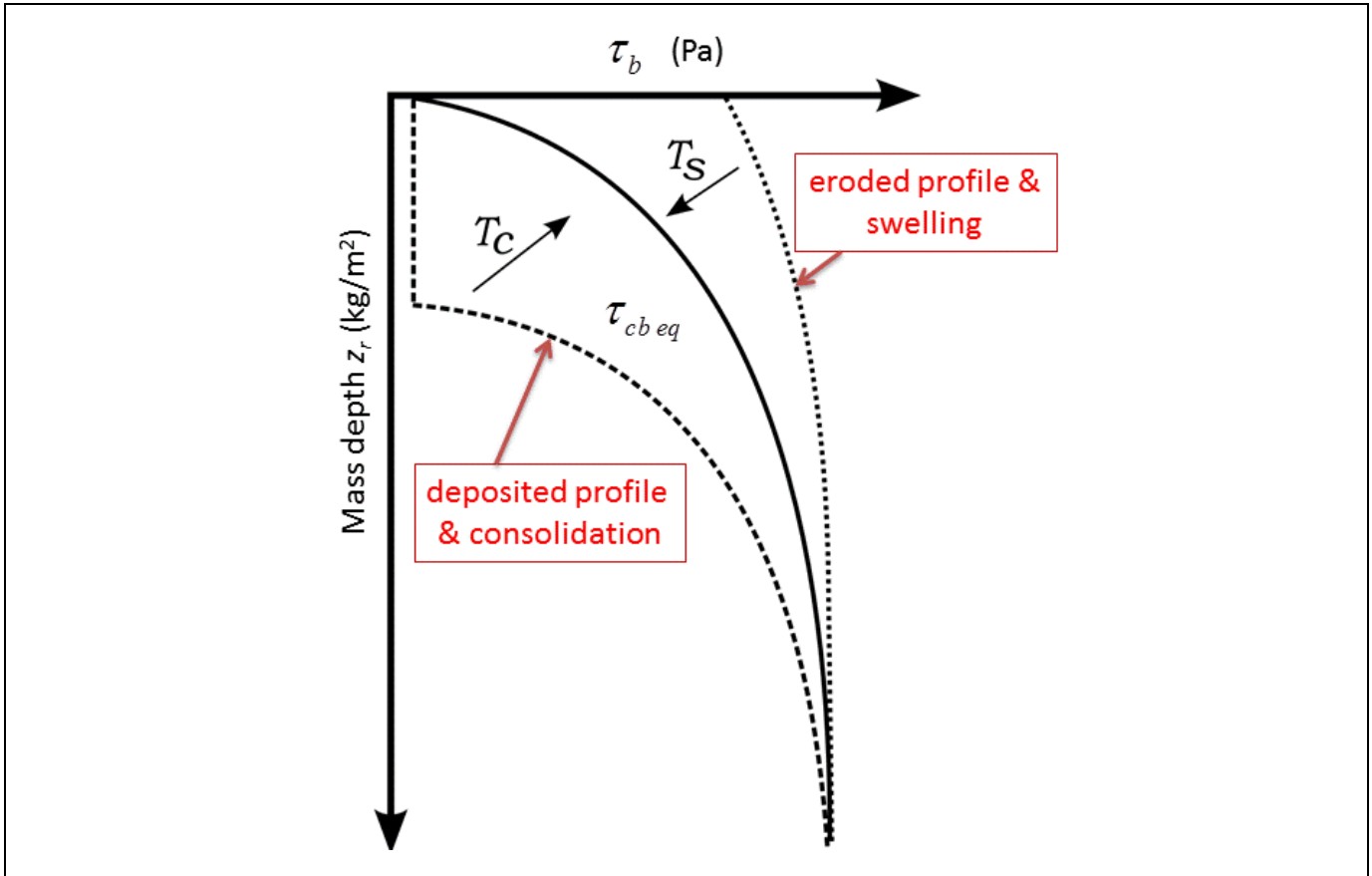

Figure 1. Conceptual diagram of consolidation and swelling (Rinehimer et al., 2008). The equilibrium bulk critical stress for erosion profile, $\tau_{cbeq}(z)$ is shown as the solid line. The dotted line represents a critical shear stress profile following sediment erosion. The dashed line is a profile after deposition of sediment with a low $\tau_c$ at the surface. The arrows indicate consolidation and swelling toward the equilibrium profile with the timescales $T_c$ and $T_s$, respectively.



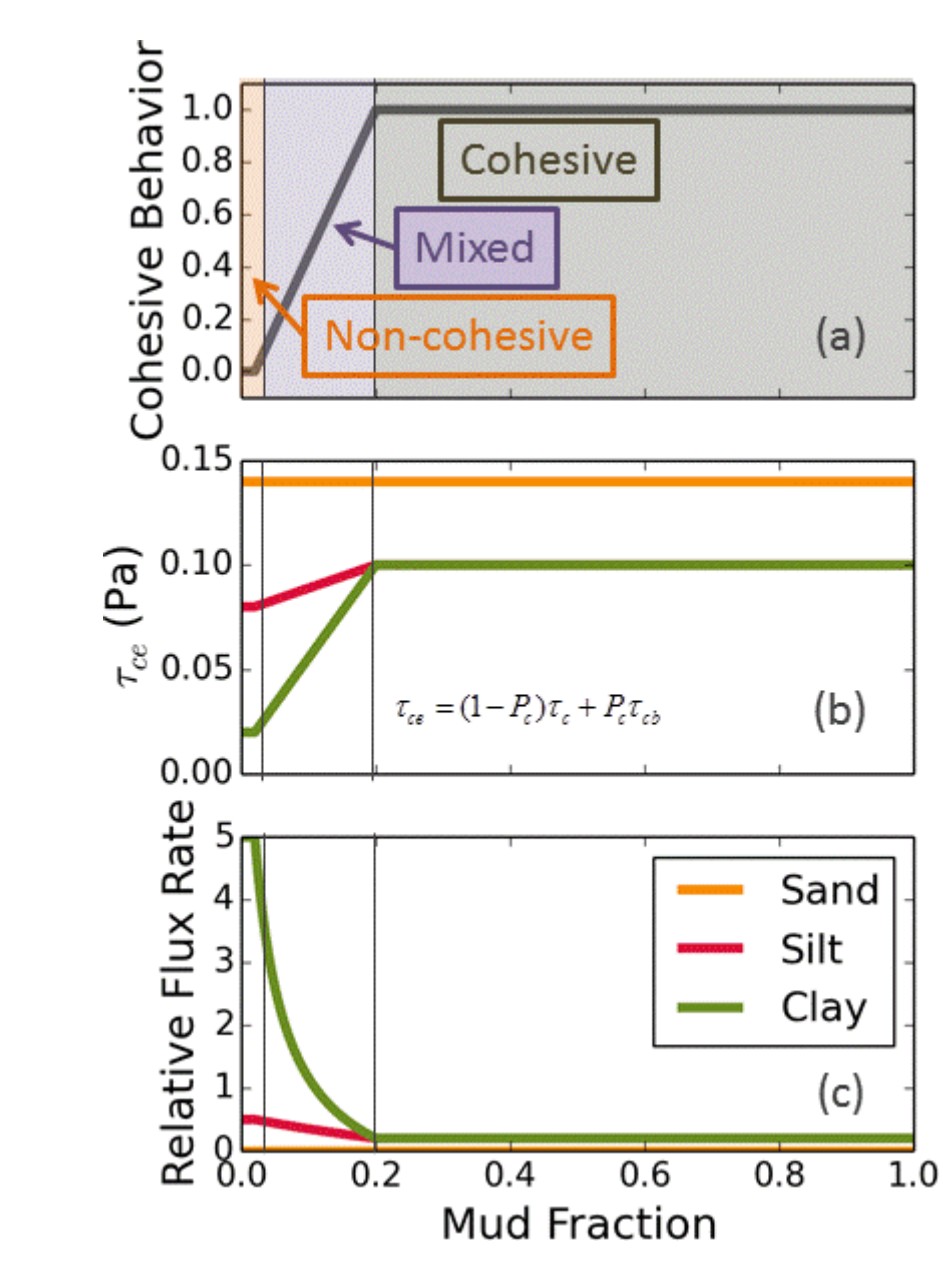

**Figure 2. Summary of mixed-bed behavior with increasing of mud fraction $f_c$ (the combined mass fraction of material in cohesive classes). (a) Cohesive behavior parameter $P_c$ as a function of $f_c$. (b) Effective critical shear stress $\tau_{ce}$ for size classes where bulk critical shear stress of the bed $\tau_{cb}$ = 0.1 Pa . (c) Relative flux (normalized excess shear stress) from the bed when bed stresses are $\sim\tau_b$ = 0.12 Pa (greater than $\tau_c$ for clay and silt primary particles, but less than $\tau_c$ for sand)**

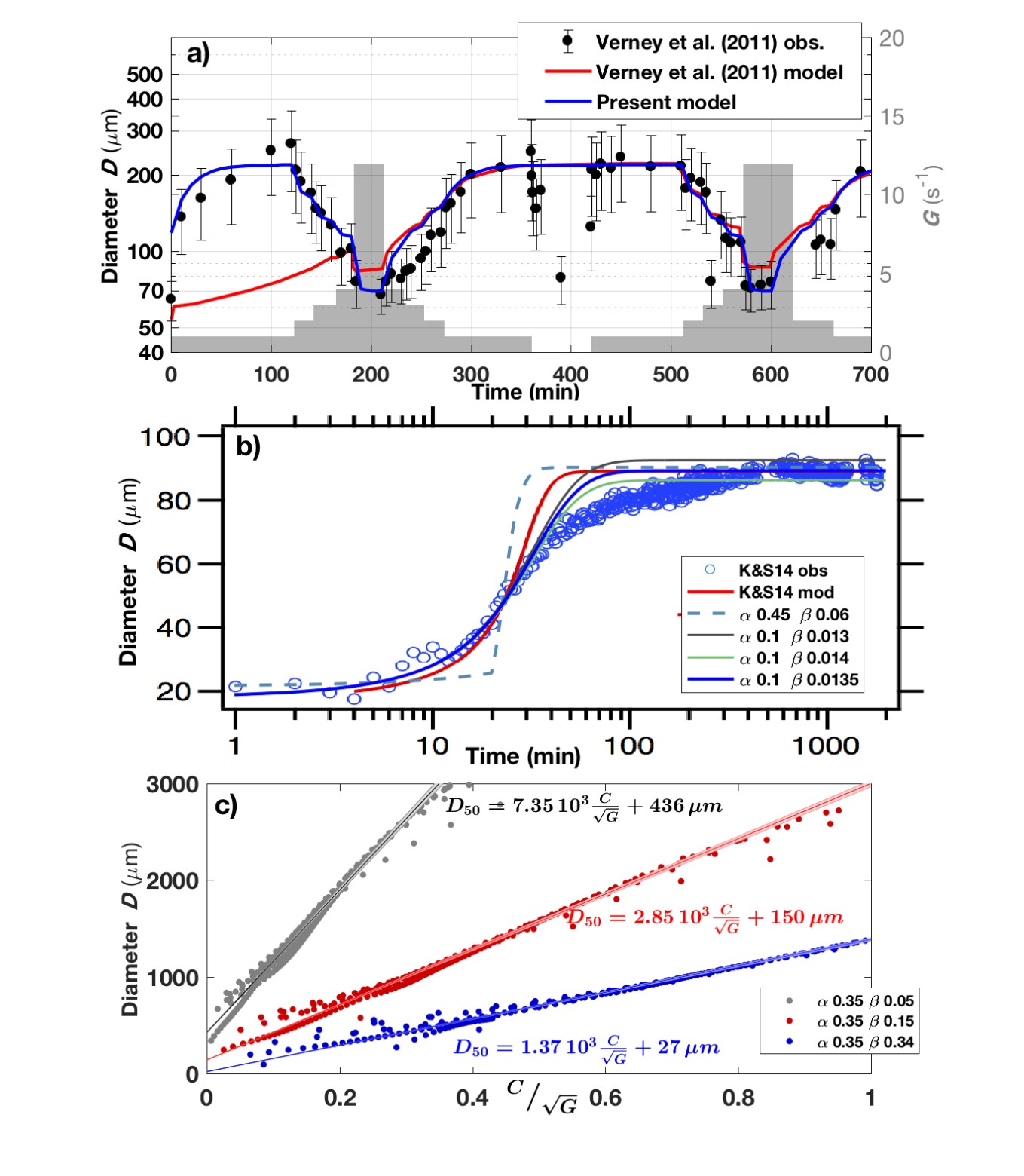

**Figure 3. Comparison of ROMS implementation of FLOCMOD with laboratory and theoretical results. (a) Laboratory response of floc size to simulated fluctuations in shear rate $G$ (gray shading) showing observed area-weighted mean floc diameter $D$ (black dots with +/ one standard deviation bars), model results presented in Verney et al., (2011; red line), and ROMS FLOCMOD simulation (blue line). (b) Laboratory response of floc size to rapid increase in shear rate from $G=0$ to $G=15$ s$^{-1}$ showing sizes measured by Keyvani and Strom (2014; K&S14; blue circles), K&S14 model results (red line), and ROMS FLOCMOD results for various combinations of aggregation and breakup parameters (dashed and colored lines). (c) Equilibrium diameters produced by steady ROMS FLOCMOD simulations with a range of concentrations, shear rates, and aggregation and breakup parameters (dots). These fall along lines with slopes determined by the ratio of aggregation and breakup parameters, according to theory (Winterwerp, 1998).**



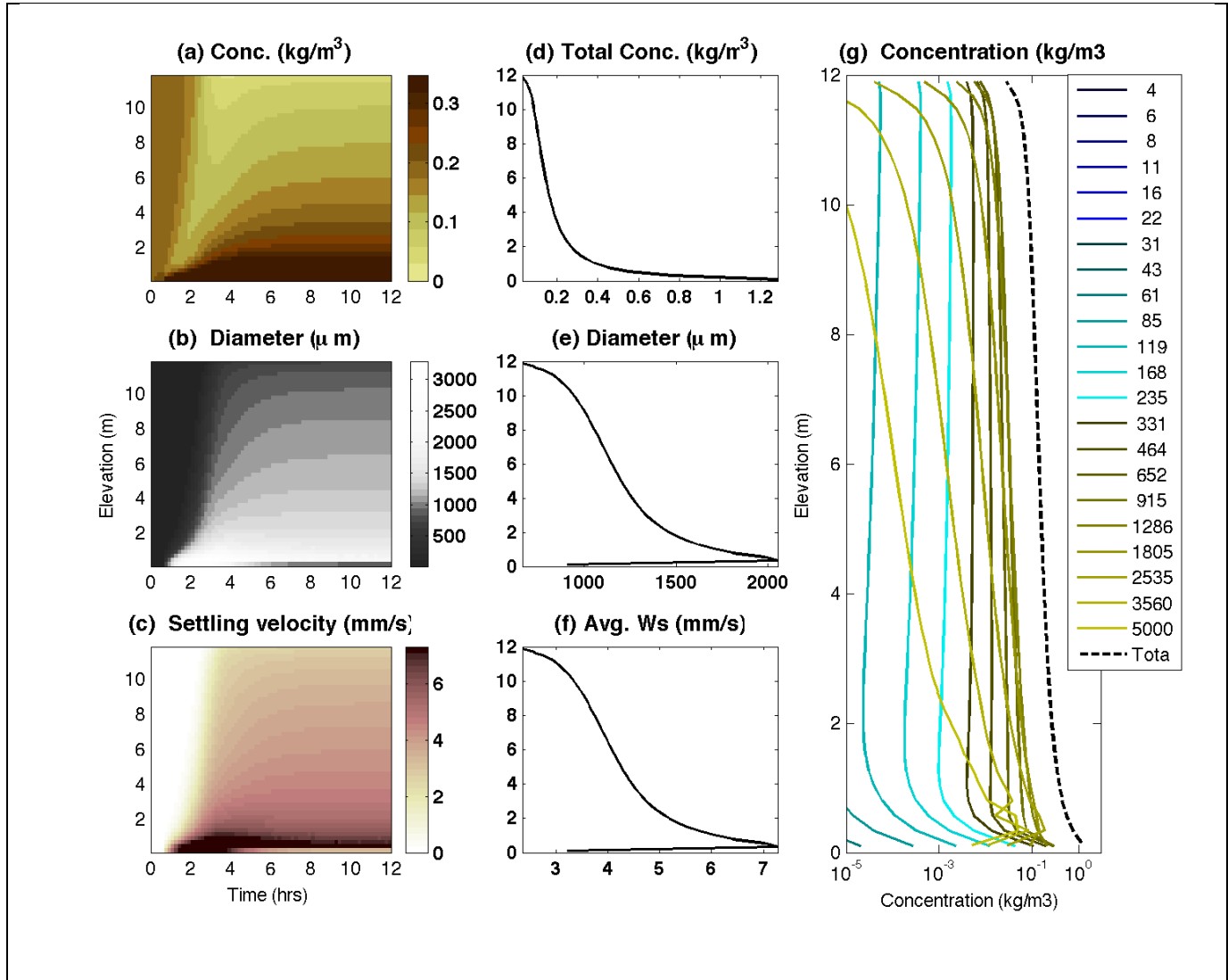

**Figure 4. Simulation of steady open-channel flow initialized with a vertically uniform concentration of 0.2 kg/m³ in the 8-μm class. Temporal evolution of profiles of (a) mass concentration (b) mass-weighted diameter (c) mass-weighted settling velocity. Final profiles of (d) concentration, (e) diameter and (f) settling velocity, and (g) final concentration profiles for each class size (colored lines) and sum of all classes (dashed line).**




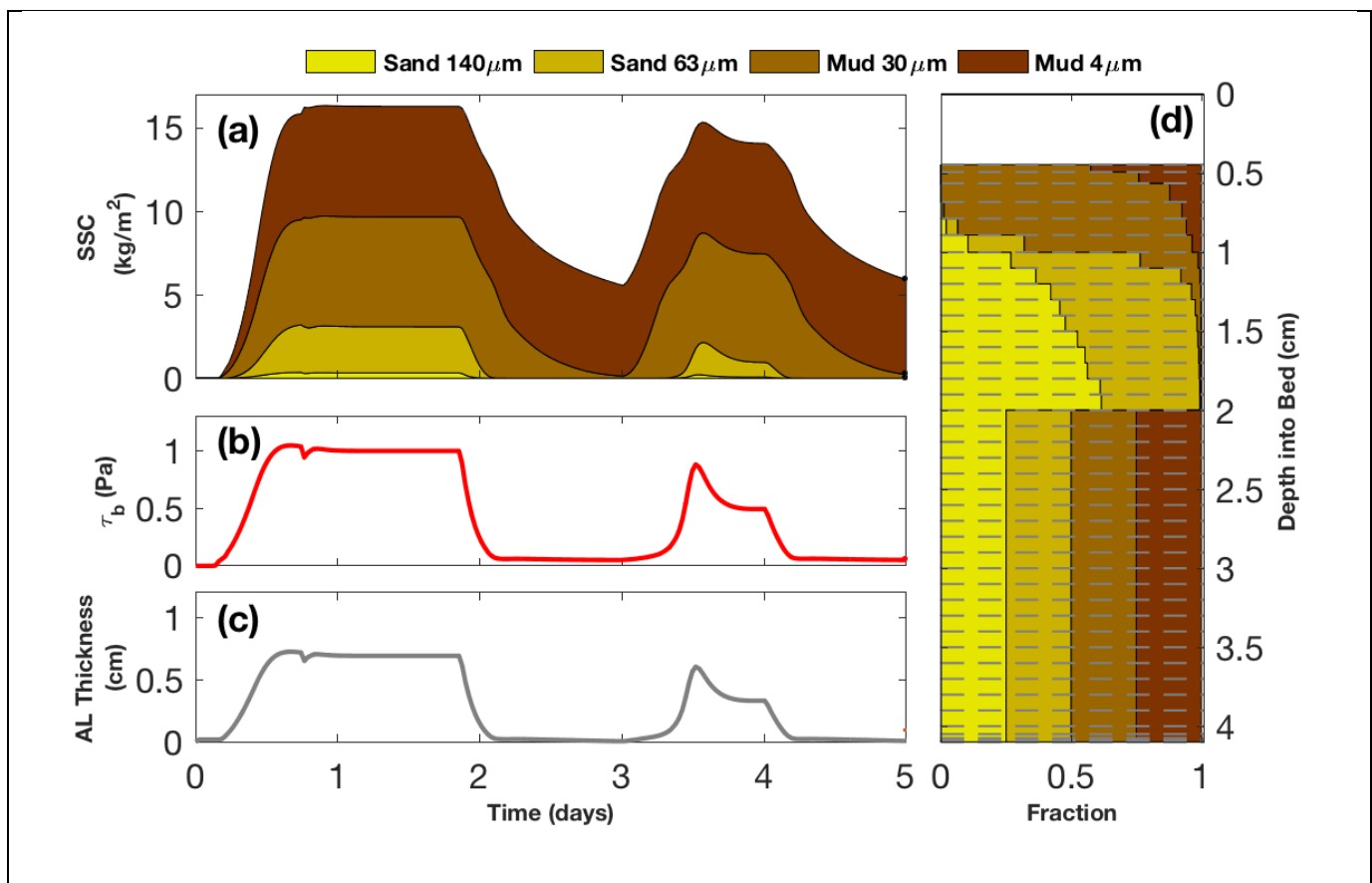

**Figure 5. Summary of the double resuspension experiment with non-cohesive sediment over 5 days. The model setup included 41 bed layers, a minimum new layer thickness of 1 mm, and four non-cohesive classes. The top horizontal panel (a) shows the time evolution of the mass of sediment in suspension, colored by size class. The middle horizontal panel (b) is the time series of bottom stress, and the bottom horizontal panel (c) shows the corresponding time series of active-layer thickness. The right panel (d) depicts the final stratigraphy relative to the initial bed level at zero and shows the fraction of each sediment class in each bed layer.**


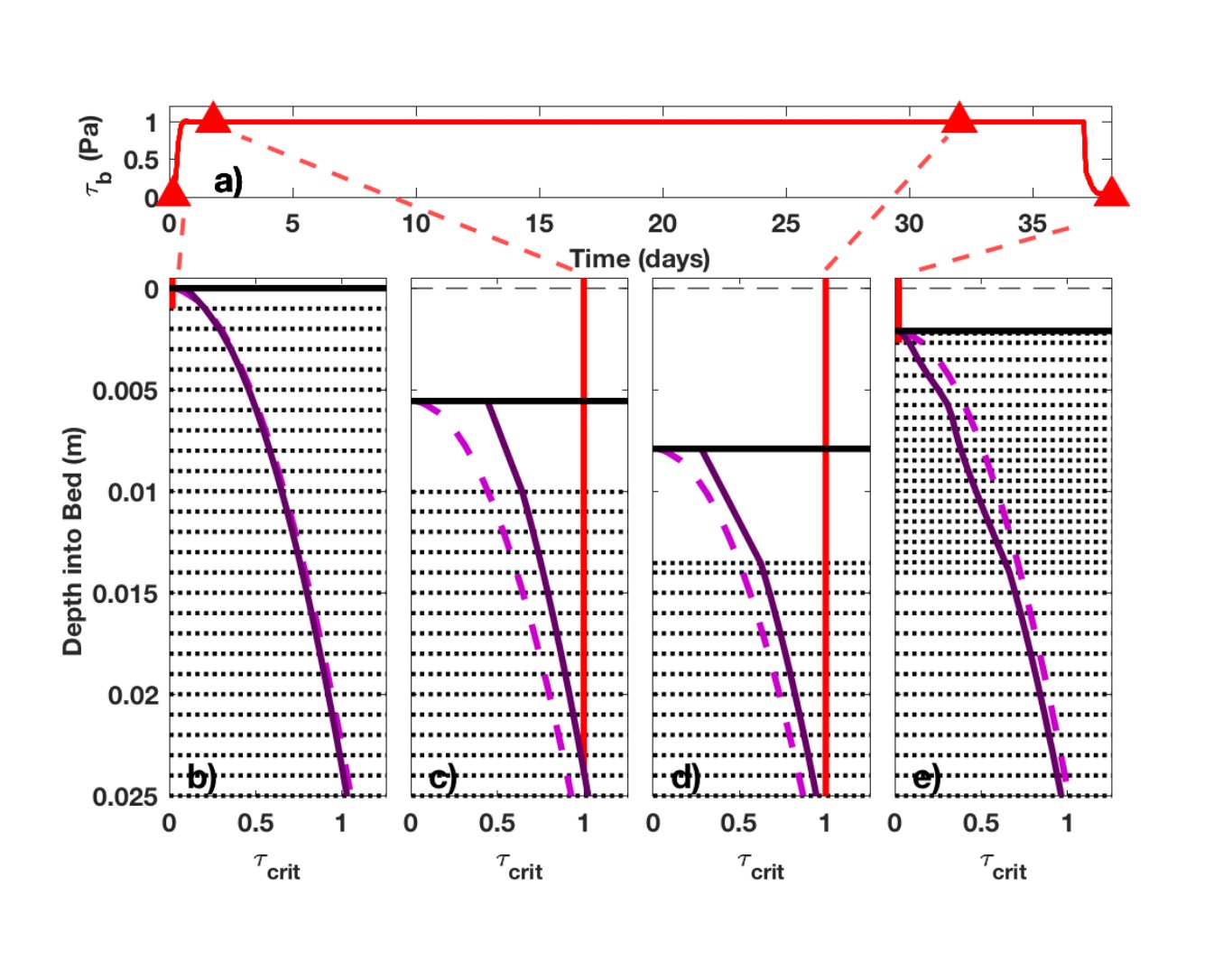

**Figure 6.** Time series of bottom stress (a) and profiles of critical shear stress for erosion during four distinct conditions: (b) initial bed condition; (c) eroded bed (after 1.3 days with $\tau_b = 1.0$ Pa); (d) after slow but continuous erosion and reduced bulk critical stress profile due to swelling after 30 days more with $\tau_b = 1.0$ Pa); and (e) rapid deposition after a day of low stress with $\tau_b = 0.1$ Pa). In the lower panels, the solid red line is the magnitude of the bottom stress ($\tau_b$), the dashed magenta line is the equilibrium profile of bulk critical stress for erosion $\tau_{cb}(z)$, and the solid purple line is the instantaneous profile of bulk critical stress for erosion. The solid black line is the instantaneous position of the top of the bed at each time, with the initial bed elevation starting at zero.

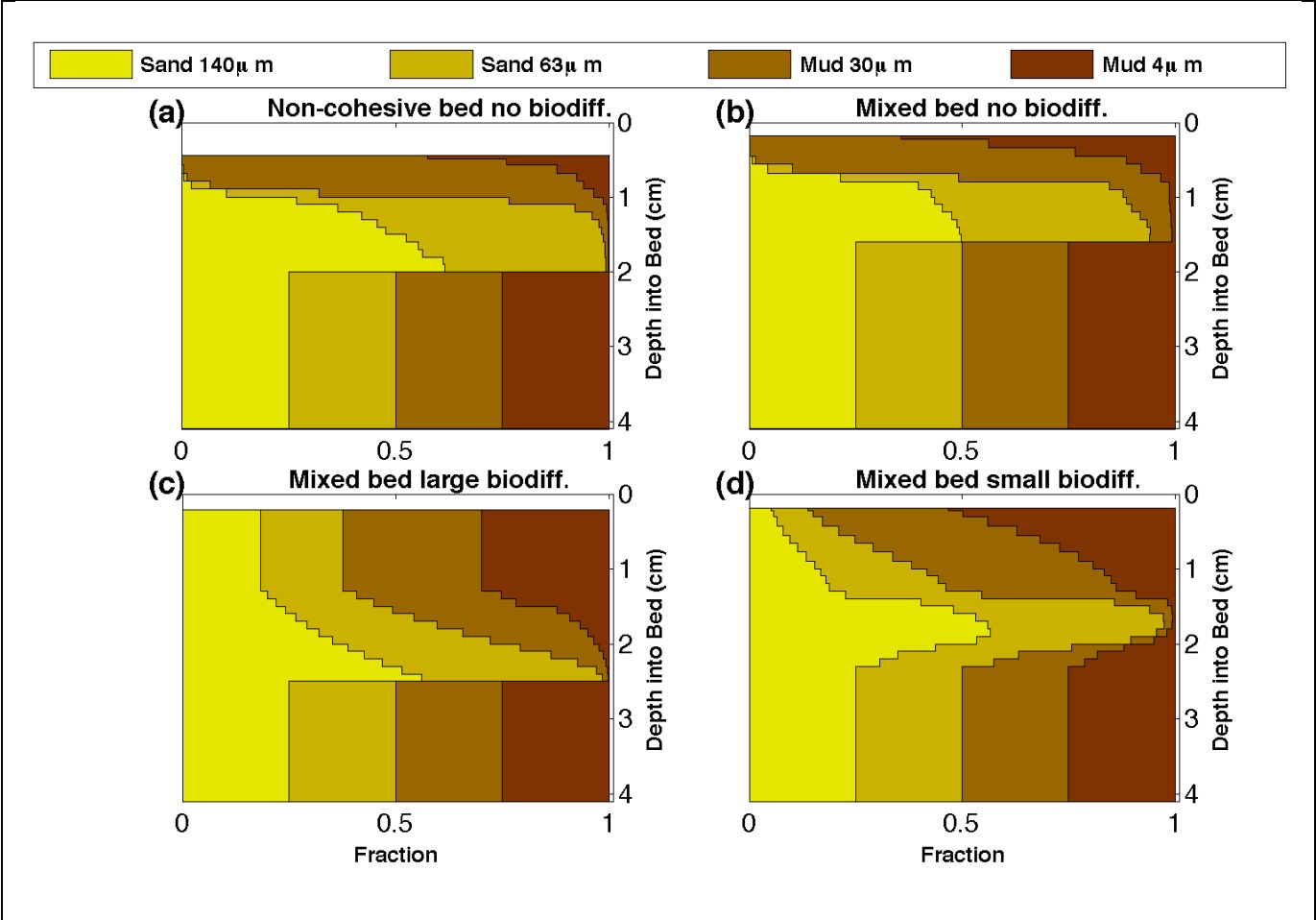

**Figure 7. Comparison of final bed stratigraphy for resuspension and settling simulations showing the fraction of each sediment class distributed in each bed layer. (a) non-cohesive bed with no biodiffusion (same as Figure 5d, included for comparison); (b) mixed bed with no biodiffusion; (c) mixed bed with large biodiffusion ($D_s=10^{-5}$ $m^2 s^{-1}$); and (d) mixed bed with small biodiffusion ($D_s=10^{-10}$ $m^2 s^{-1}$). The final sediment fraction distribution after two successive erosion/deposition events lasting five days (similar to Figure 5b) is shown. The same four sediment classes were used in all experiments, but their cohesive behavior varied.**

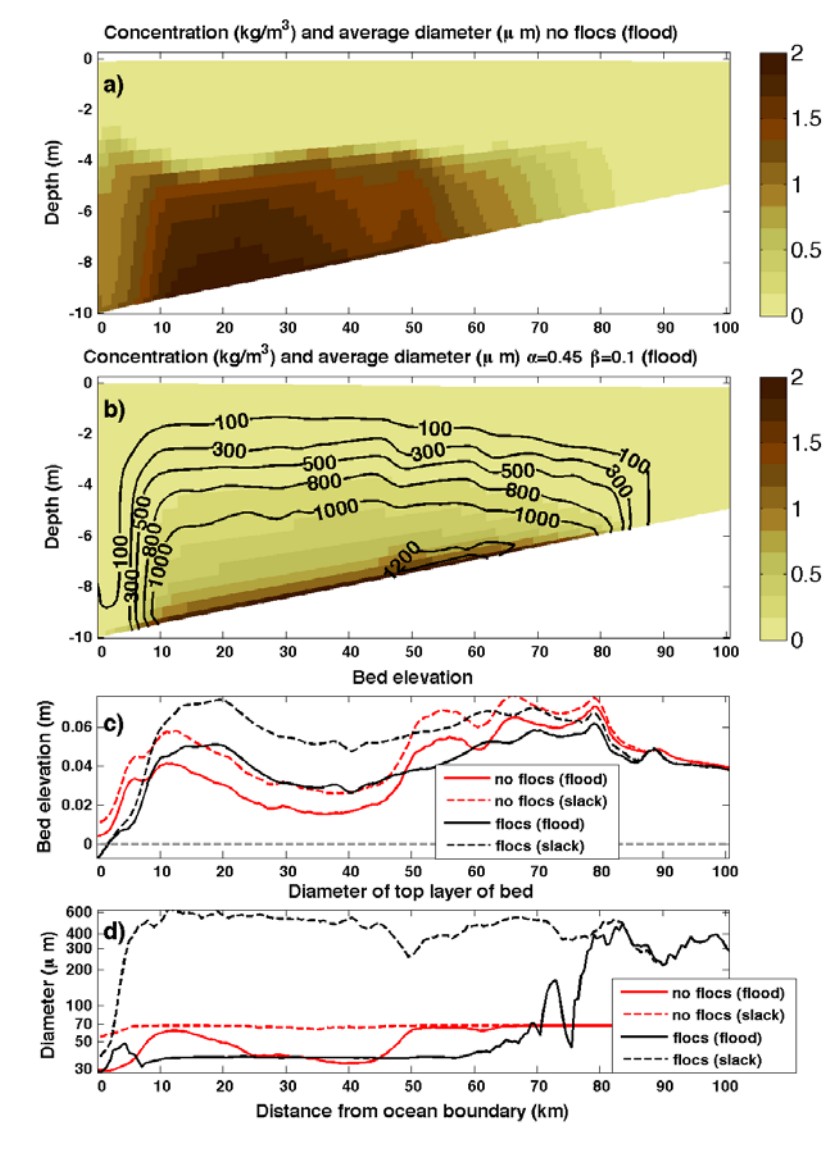

**Figure 8. Comparison of estuarine turbidity maxima simulations with and without floc dynamics. a) Two-dimensional (along-estuary and vertical) snapshot of suspended particle concentrations (shaded) without floc dynamics near the end of flood tide. All of the suspended material was in the 37-μm class. b) Snapshot of suspended particle concentrations at the same time in the simulation, but with simulated floc dynamics (shading), overlain by contours of mean particle diameters. c) Along-estuary profiles of bed elevations for simulations without floc dynamics (red) and with floc dynamics (black) at the peak of flood tide (solid lines) and at post-flood slack tide (dashed lines). d) Along-estuary profiles of mean particle diameter in the top layer of the seabed, using the same notation as (c).**


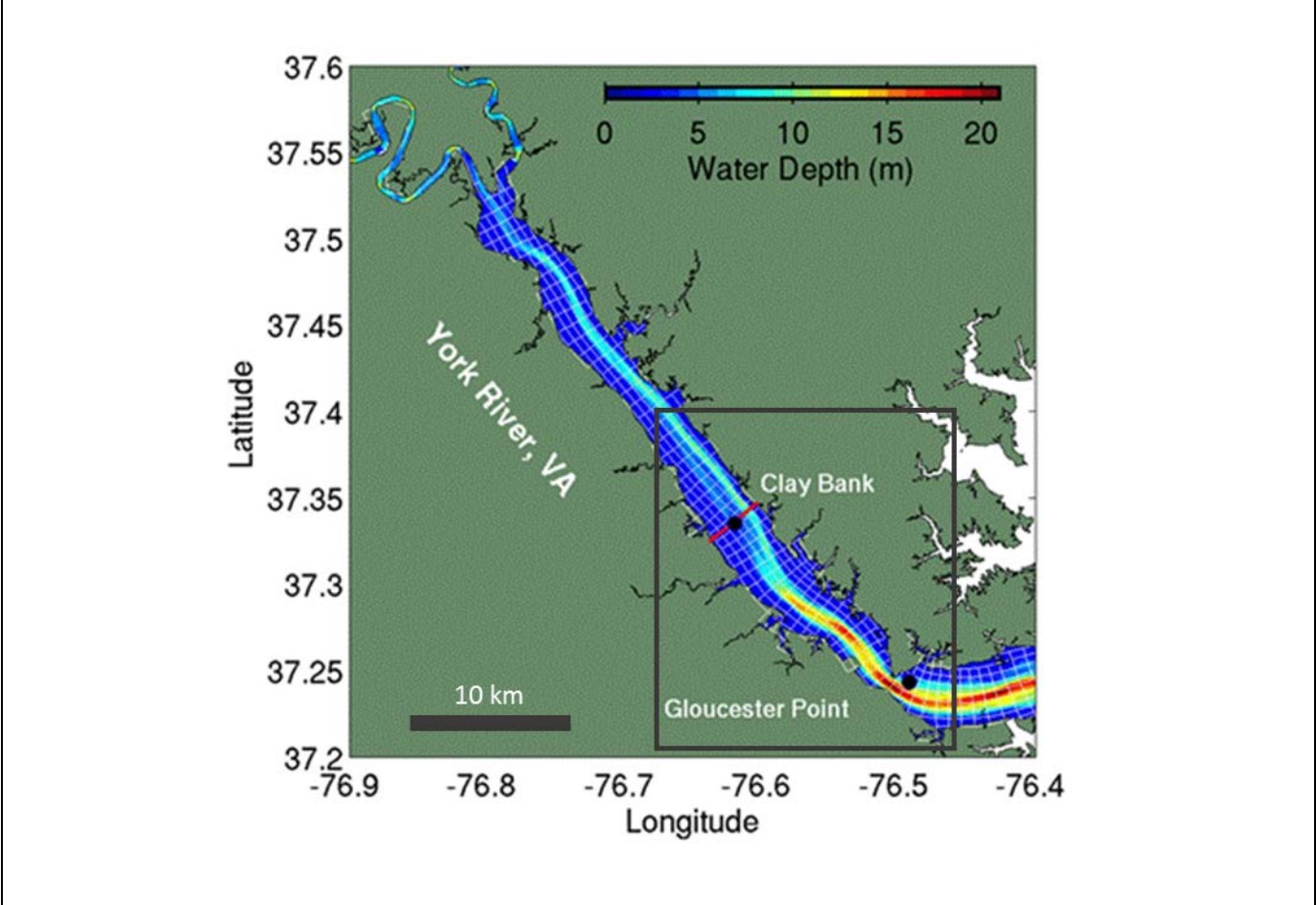

**Figure 9. York River bathymetry (color scale), and model grid (white lines show every fifth grid line in the along- and across-channel directions). The region outlined in grey is expanded in Figure 10.**




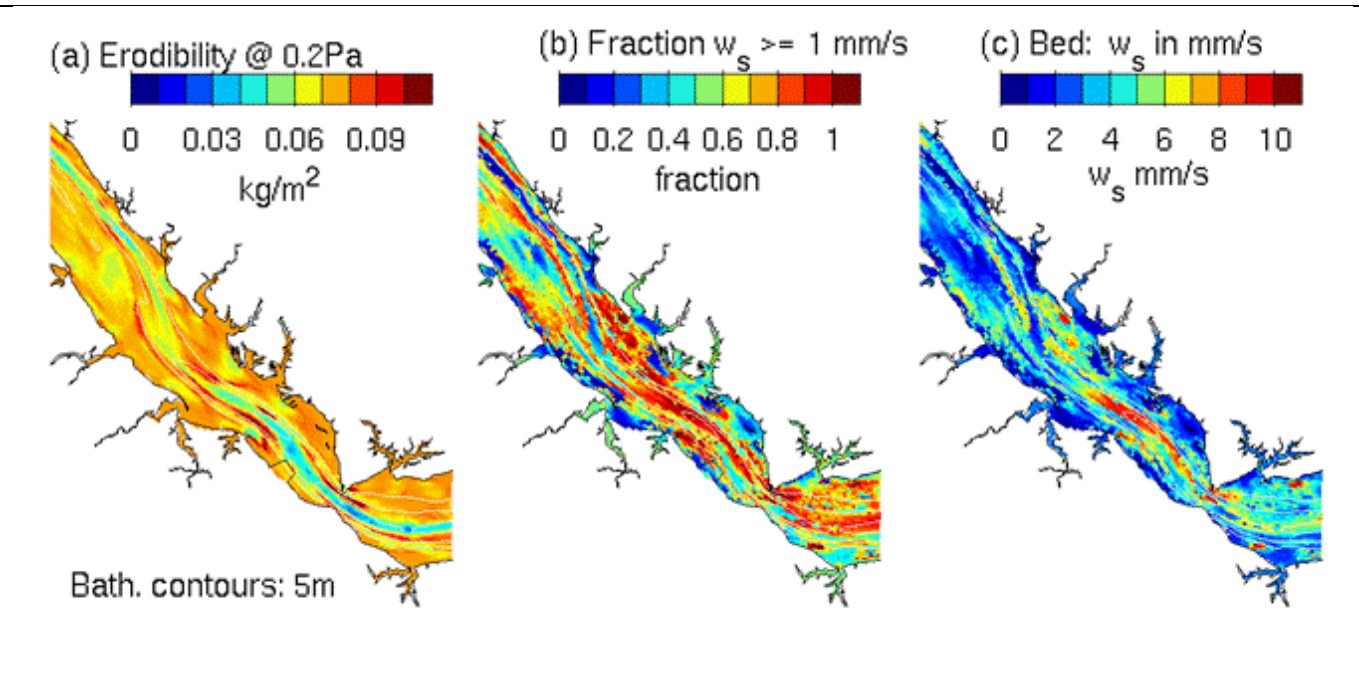

**Figure 10. Model estimates of seabed properties after two months of tidal forcing and constant, average freshwater discharge. (a) Erodibility of the seabed, calculated as the thickness of the layer having a critical shear stress exceeded by 0.2 Pa. (b) Fraction of the surficial sediment in the "faster settling" size class. (c) Average settling velocity of surficial sediment.**
