# Peer review of "Cohesive and mixed sediment in the Regional Ocean Modeling"

_Geoscientific Model Development, 2017_

## Short Comment (SC1) · 13 Dec 2017

As explained in https://www.geoscientific-model-development.net/about/manuscript_types.html any model description paper needs to have a "Code Accessibility" section. Please add this to your paper.

As also explained in this document GMD is encouraging that authors upload the program code of models (including relevant data sets) as a supplement or make the code

and data or the exact model version described in the paper accessible through a DOI (digital object identifier). In case your institution does not provide the possibility to make electronic data accessible through a DOI you may consider other providers (eg. zenodo.org of CERN) to create a DOI. Please note that in the code accessibility section you can still point the reader how to obtain the newest version.

If for some reason the code and/or data cannot be made available in this form (e.g. only via e-mail contact) the"Code Accessibility" section need to clearly state the reasons for why access is restricted (e.g. licensing reasons).

Lutz Gross GMD Executive Editor

---

## Editor Comment (EC1) · G. Munhoven (Editor) · 13 Dec 2017

This model description paper has the required "Code availability" section (not "accessibility") starting on line 644. Its is called "Code and Data Availability", with my agreement, as the required files include significant amounts of data. The licensing terms have been adequately stated (MIT/X License) and the procedure to follow to get access to the complete code - actually the complete Subversion repository - is clearly

given (send eMail to jcwarner@usgs.gov). The version of the model that is described here is revision 1179, as stated in the title and repeated in the code availability section and can thus be retrieved without ambiguity.

For the purpose of the review process, the topical editor has been given an anonymous way to access the code that he may provide the required credentials to reviewers who wish to get access to the code, while preserving their anonymity. Peers who wish to submit a comment and want to inspect the code may apply for access via the standard way, as they may not submit their comments anonymously anyway. Uploading the code as a supplement is not an option, because is exceeds the 50MB limit by an order of magnitude (at least).

————————————————————

---

## Referee Comment (RC1) · Anonymous Referee #1 · 20 Dec 2017

**Review "Cohesive and mixed sediment in the Regional Ocean Modeling System (ROMS v3.6) implemented in the Coupled Ocean Atmosphere Wave Sediment-Transport Modeling System (COAWST r1179)" by Sherwood et al., GMD Discussions**

1. Does the paper address relevant scientific modelling questions within the scope of GMD? Does the paper present a model, advances in modelling science, or a modelling protocol that is suitable for addressing relevant scientific questions within the scope of EGU?

The authors extended an existing model for regional-scale coastal sediment transport and morphodynamics by implementing a number of previously developed routines that account for cohesive sediment and biogemorphology effects. The upgraded model is most likely of interest to both academics and engineers working in the coastal community.

2. Does the paper present novel concepts, ideas, tools, or data?

The present study does not present completely new model concepts, but instead, it combines existing model formulations that were developed by the same authors in preceding studies (Warner et al., 2008; Rinehimer et al., 2008; Verney et al., 2011). This leads to an upgraded version of the ROMS model, which is considered a novel tool that is worthy of publication.

3. Does the paper represent a sufficiently substantial advance in modelling science?

Yes.

4. Are the methods and assumptions valid and clearly outlined?

The implemented methods have been described in preceding studies and seem valid. However, some components in the model and underlying assumptions require additional clarification, see my specific remarks below.

5. Are the results sufficient to support the interpretations and conclusions?

The authors present results of a number of idealized "demonstration cases" and a realistic application. These cases are generally interesting and the results support the conclusions. Specific remarks regarding the simulations and the interpretations of results are listed below.

6. Is the description sufficiently complete and precise to allow their reproduction by fellow scientists (traceability of results)? In the case of model description papers, it should in theory be possible for an independent scientist to construct a model that, while not necessarily numerically identical, will produce scientifically equivalent results. Model development papers should be similarly reproducible. For MIP and benchmarking papers, it should be possible for the protocol to be precisely reproduced for an independent model. Descriptions of numerical advances should be precisely reproducible.

The explanations are at some points rather short, and for a full understanding of the methodology (e.g. equations and numerical implementation) the reader has to turn to preceding papers by these authors and to information contained in the Supplement. I appreciate, however, that a journal format may not allow to fully explain all the details. Given that the numeric code is available to anyone, and that the community is explicitly invited to use the code, I expect the present work to be fully reproducible.

7. Do the authors give proper credit to related work and clearly indicate their own new/original contribution?

Yes.

8. Does the title clearly reflect the contents of the paper? The model name and number should be included in papers that deal with only one model.

Yes.

9. Does the abstract provide a concise and complete summary?

Yes.

10. Is the overall presentation well structured and clear?

Yes.

11. Is the language fluent and precise?

The paper is well-written in fluent English.

12. Are mathematical formulae, symbols, abbreviations, and units correctly defined and used?

Yes.

13. Should any parts of the paper (text, formulae, figures, tables) be clarified, reduced, combined, or eliminated?

Yes, see specific comments below.

14. Are the number and quality of references appropriate?

Yes.

15. Is the amount and quality of supplementary material appropriate? For model description papers, authors are strongly encouraged to submit supplementary material containing the model code and a user manual. For development, technical, and benchmarking papers, the submission of code to perform calculations described in the text is strongly encouraged.

The 27-page Supplement provides details on the implemented methodology, including a description of the main equations. The code is not directly provided but is available upon request.

**Specific major comments**

1. Given that one of the model goals is to simulate morphologic change (Line 98), I am surprised that the realistic application of the model to the York River Estuary does not address the morphologic evolution at all. Is the model also capable of accurately simulating longer-term morphologic changes in a complex environment such as an estuary? If the authors were to run the model for a longer simulation time (say a few years), would the model reproduce a reliable evolution of the main geomorphologic features (banks, creeks, shoals, ...) of the estuary? To me this is a key issue in trusting the model's performance, and results or a general discussion on this issue are essential.
It would also be interesting to see how the modeled morphology would differ for simulations with the present, upgraded model, relative to simulations with the original model by Warner et al. (2008).

2. A topic that is overlooked, or at least not considered in the manuscript, is bedload transport - apart from a general notion that the stratigraphy is relevant for bedload transport (L.207). This is rather confusing and I believe the following topics should be addressed:
   A. Is size-selective bedload transport included at all? If yes, which model is used?
   B. How does the bedload transport depend on the particle size distribution in e.g. the active layer?
   C. How is the critical bed shear stress for bedload determined? Is the applied method consistent with the methodology proposed for the erosion rate in Section 2.4?

**Minor comments**

Section 2: While a section is devoted to the flux into the bed (2.2.1), the erosive flux from the bed into suspension is not described at all. The method and equations used to calculate the erosive flux should be added.

L198-201: It is not instantly clear how the floc size changes in the bed. Deflocculation (L.199) suggests (to me) that flocs degrade to loose sediment particles, but this appears to be in contrast with the preceding statement ("flocs erode as denser, more angular aggregates"). Reading further (and checking the Appendix), I understand that the cohesive size classes tend to an equilibrium distribution, which means that the reverse may also happen: loose clay/silt grains that form aggregates in the bed. Therefore I believe the term "deflocculation" is not well-chosen for this process.

L206-221: What happens when the bed is emerged? Are processes like shrinking/swelling accounted for in the bed stratigraphy module, or can these be added in future? Drained clay soils will become more compacted, which is accounted for in the empirical method for the critical bed shear stress. However, are these processes also considered relevant for the determination of the bed layers?

Section 2.4 The method to quantify tau_cb is rather crude. Could the approach be somehow improved by taking the information of the floc size distribution in the bed (Section 2.2) into account? Any reflection and/or suggestions to improve this approach would be useful.

L269: The explanations related to P_c are difficult to follow. Insertion of equation S29 from the Supplement would help understanding this section.

Section 3: The demonstration cases in Sections 3.1 and 3.2 are very interesting and insightful.

L318-325: More explanation regarding the Verney et al. (2011) experiment would be useful. For instance, what is the time of one full cycle in the experiment? Is the dip in the measurements around $t = 400$ min due to periodicity in velocity forcing, and why doesn't the model reproduce this dip?

L330 introduces the aggregation/collision parameter alfa and break-up/fragmentation parameter beta. Overlooking all test cases in Section 3, alfa varies by a factor 5 and beta by a factor 10. Results appear to be quite sensitive (see e.g. Fig. 3b-c) to the values for alfa and beta. How do values for alfa and beta relate to the physical properties of a cohesive mixture? And how can users determine the optimum value for these parameters? To what extent are the values used for the simulations in Fig.3b accurate (beta <0.02), as they deviate strongly from beta values for the other simulations in the manuscript?

L430: The active layer is defined as the upper-most layer (L222) which I interpret as being a single grid cell. Consequently I find the explanation in L430 somewhat confusing ("the active layer ... extended 2 cm below the surface") given that one grid cell is 1 mm. Can the active layer comprise multiple cells/layers that erode at once, or is the 2cm erosion explained by a stepwise removal of the top "active" layer in 20 time steps?

L460 "compare Figures 6c, d": I understand what I should be seeing, but the differences between the curves are too small to detect them by eye. Perhaps the period with high bed-stress should be extended to make the point.

Fig. 3a: what do the error bars depict? 95% C-I, or +/- 1*st.dev?

**Technical corrections**

L78-79: "that that"

L104: "seagrass growth model" → models?

L335 full stop missing at end of sentence.

L513 last sentence refers to Figure 8a, but no information on the grain size is contained in this figure. Consequently also the title of Fig. 8a is incorrect.

---

## Referee Comment (RC2) · Anonymous Referee #2 · 24 Jan 2018

The authors present the implementation of a cohesive and mixed sediment module within the COAWST (ROMS based system). They provide a thorough and extensive framework that includes floc model, stratigraphy and bed mixing, critical stress for erosion of cohesive sediment. None of the individual components is particularly novel in isolation, but the overall model combining all aspects does present a significant advance in coastal sediment transport modelling. The manuscript is well written and I enjoyed reading it. There are a few issues that would need to be addressed in a revision.

The most important issue is that it is not clear how the floc model is combined with the vertical ROMS grid and vertical sediment fluxes (turbulent suspension and settling) to determine suspensions of cohesive sediments. Are these actually included (the steady state test suggests yes but the comparison to Verney (2011) no)? The key discrepancy in the model-data comparison in figure 3a at t=400 min corresponds to a settling stage. In Verney et al. (2011), the settling dip was not reproduced either as particle deposition was not allowed in the 0D model. Is the same explanation also valid here?

Another weakness is that, even though the manuscript includes a number of test cases, it looks to me that there is a lack of validation. Only the floc model is validated against measurements and there is no validation against field observations, especially for cohesive suspended sediments. This is somewhat frustrating and looks like a missed opportunity as LISST instruments are now relatively commonly deployed in the field. Since they measure concentrations for a number of floc size classes, they would appear to be well suited to provide datasets for validation and model-observation comparisons.

Given that the new algorithms are incorporated in COAWST, I am wondering about coupling and/or compatibility with the wave module(s). While a full test of this may be outside of the scope of the paper, I think discussing this point would strengthen the manuscript.

Specific comments:

Section 2.2.1: I'm not sure whether this is the best place to present fluxes into the bed. The alternative (which probably would be my preference) is to combine with erosion into a "bed water column exchange" section.

Figure 3a,b: It would be helpful to also have the temporal evolution of G shown. Since the authors include the modelling results of Verney et al. (2011), it would be useful to explain the reason for the different model results during the first aggregation stage

(initial distribution), instead of relying on the reader checking in Verney et al. (2011).

Section 2.6: The new modules are added to the existing sediment transport model in ROMS (Warner et al., 2008) and in COAWST, which includes waves. The presence of bedforms and waves may induce pressure gradients at the sediment bed, which would in turn induce interstitial porewater flow in the bed. This process can entrain fine particles into a coarser sediment bed (e.g., Huettel et al., 1996, Limnol. Oceanogr., 41(2), 1996, 309-322). It would be welcome for the authors to comment on this process and its inclusion (or not) in the present framework.

Figure 4: there appears to be a "kink" in the concentration for one specific profile (3560 microns?). What is the cause?

Figure 8: Caption should include details on what the different panels (a, b, c, d) show.

Technical corrections:

Line 79: one too many that

Line 145-146 vs lines 115-116: Repetition, please remove one of the two.

---

## Short Comment (SC2) · 15 Mar 2018

Response to interactive comment by Anonymous Referee #1 on "Cohesive and mixed sediment in the Regional Ocean Modeling System (ROMS v3.6) implemented in the Coupled Ocean Atmosphere Wave Sediment-Transport Modeling System (COAWST r1179)" by Christopher R. Sherwood et al. Comment received 20 December, 2017.

The authors thank Anonymous Referee #1 for detailed and insightful comments on our manuscript. Here, we respond to those comments and indicate changes we have made in the manuscript to address them. Comments are reproduced in ***bold+italics***; our response is in plain text.

***1. Does the paper address relevant scientific modelling questions within the scope of GMD? Does the paper present a model, advances in modelling science, or a modelling protocol that is suitable for addressing relevant scientific questions within the scope of EGU?***

***The authors extended an existing model for regional-scale coastal sediment transport and morphodynamics by implementing a number of previously developed routines that account for cohesive sediment and biogemorphology effects. The upgraded model is most likely of interest to both academics and engineers working in the coastal community.***

***2. Does the paper present novel concepts, ideas, tools, or data?***

***The present study does not present completely new model concepts, but instead, it combines existing model formulations that were developed by the same authors in preceding studies (Warner et al., 2008; Rinehimer et al., 2008; Verney et al., 2011). This leads to an upgraded version of the ROMS model, which is considered a novel tool that is worthy of publication.***

***3. Does the paper represent a sufficiently substantial advance in modelling science?***

***Yes.***

***4. Are the methods and assumptions valid and clearly outlined?***

***The implemented methods have been described in preceding studies and seem valid. However, some components in the model and underlying assumptions require additional clarification, see my specific remarks below.***

***5. Are the results sufficient to support the interpretations and conclusions?***

***The authors present results of a number of idealized "demonstration cases" and a realistic application. These cases are generally interesting and the results support the conclusions. Specific remarks regarding the simulations and the interpretations of results are listed below.***

***6. Is the description sufficiently complete and precise to allow their reproduction by fellow scientists (traceability of results)? In the case of model description papers, it should in theory be possible for an independent scientist to construct a model that, while not necessarily numerically identical, will produce scientifically equivalent results. Model development papers should be similarly reproducible. For MIP and benchmarking papers, it should be possible for the protocol to be precisely reproduced for an independent model. Descriptions of numerical advances should be precisely reproducible.***

*The explanations are at some points rather short, and for a full understanding of the methodology (e.g. equations and numerical implementation) the reader has to turn to preceding papers by these authors and to information contained in the Supplement. I appreciate, however, that a journal format may not allow to fully explain all the details. Given that the numeric code is available to anyone, and that the community is explicitly invited to use the code, I expect the present work to be fully reproducible.*

*7. Do the authors give proper credit to related work and clearly indicate their own new/original contribution?*

*Yes.*

*8. Does the title clearly reflect the contents of the paper? The model name and number should be included in papers that deal with only one model.*

*Yes.*

*9. Does the abstract provide a concise and complete summary?*

*Yes.*

*10. Is the overall presentation well structured and clear?*

*Yes.*

*11. Is the language fluent and precise?*

*The paper is well-written in fluent English.*

*12. Are mathematical formulae, symbols, abbreviations, and units correctly defined and used?*

*Yes.*

*13. Should any parts of the paper (text, formulae, figures, tables) be clarified, reduced, combined, or eliminated?*

*Yes, see specific comments below.*

*14. Are the number and quality of references appropriate?*

*Yes.*

*15. Is the amount and quality of supplementary material appropriate? For model description papers, authors are strongly encouraged to submit supplementary material containing the model code and a user manual. For development, technical, and benchmarking papers, the submission of code to perform calculations described in the text is strongly encouraged.*

*The 27-page Supplement provides details on the implemented methodology, including a description of the main equations. The code is not directly provided but is available upon request.*

Thank you for this comprehensive review.

*Specific major comments*

*1. Given that one of the model goals is to simulate morphologic change (Line 98), I am surprised that the realistic application of the model to the York River Estuary does not address the morphologic evolution at all. Is the model also capable of accurately simulating longer-term morphologic changes in a complex environment such as an estuary? If the authors were to run the model for a longer simulation time (say a few years), would the model reproduce a reliable evolution of the main geomorphologic features (banks, creeks, shoals, …) of the estuary? To me this is a key issue in trusting the model's performance, and results or a general discussion on this issue are essential.*

*It would also be interesting to see how the modeled morphology would differ for simulations with the present, upgraded model, relative to simulations with the original model by Warner et al. (2008).*

We agree that validation of the model for long simulations of geomorphological evolution is needed. And we admit that it will be a challenge to match observations of geomorphological change in cohesive environments, and can't affirm that the model will reliably reproduce changes in banks, creeks, or shoals. But, as we responded to Referee #2, validation of each component of the model would substantially expand the scope of an already lengthy paper. Comparisons of each component of the model with field data would require introduction of the observations and analysis of the inevitable discrepancies between the model and data. The goal of the paper is to describe the modeling methods, and we hope that our demonstrations of potential applications, which produce plausible results, provide sufficient guidance and incentive for others to apply and evaluate the model. We look forward to doing so ourselves. We have not changed the manuscript to address this comment.

*2. A topic that is overlooked, or at least not considered in the manuscript, is bedload transport - apart from a general notion that the stratigraphy is relevant for bedload transport (L.207). This is rather confusing and I believe the following topics should be addressed:*

*A. Is size-selective bedload transport included at all? If yes, which model is used?*

*B. How does the bedload transport depend on the particle size distribution in e.g. the active layer?*

*C. How is the critical bed shear stress for bedload determined? Is the applied method consistent with the methodology proposed for the erosion rate in Section 2.4?*

A. Yes. The CSTMS bedload transport equations included in ROMS (Warner et al., 2008) are available and suitable for transporting the non-cohesive components in a mixed bed simulation. There are presently two options: the Meyer-Peter Mueller equation, or the Soulsby equations that include asymmetric transport by waves. The transport rates are size dependent, as discussed below.

B. These equations use the user-specified particle critical shear stress for erosion for each size class, and act on any non-cohesive classes present in the top (active) layer when Tb exceeds Tcrit for that size class AND Tau_b > Tau_cb when mixed sediment is present. In other words, a sand grain embedded in a cohesive bed will not move unless the bed stress is both greater than the bulk critical shear stress of the bed and the particle critical shear stress needed to mobilize the sand grain. We assume that cohesive sediment does not undergo bedload transport; eroded cohesive material goes directly into suspension.

C. The critical bed shear stress for bedload in a mixed bed is the critical particle shear stress computed from, for example, a Shields relationship. However, the material will not undergo bedload transport unless the bulk critical shear stress for the bed (as described in Section 2.5 [now 2.6]) is exceeded. The

erosion rate (flux from the bed into suspension) is governed by the greater of the two critical shear stress values. We don't think there is inconsistency in this approach, but it does assume that the presence of cohesive sediment does not affect the bedload transport rates of available non-cohesive sediment.

Text has been added to the end of Section 2.5 [now 2.6] as follows:

"Non-cohesive sediment classes are subject to bedload transport when the bottom stress exceeds both the bulk critical shear stress of the top (active) layer and the particle critical shear stress for that class. In these cases, the transport-rate equations still calculate bedload transport based on excess shear stress associated with the non-cohesive particle critical shear stress, as described in Warner et al (2008). Cohesive classes are not subject to bedload transport; if the bulk critical shear stress of the bed is exceeded, we assume they will go directly into suspension."

We thank the reviewer for bringing up this issue, because it led us to an error in the code that will be fixed in the release accompanying the final manuscript.

***Minor comments***

***Section 2: While a section is devoted to the flux into the bed (2.2.1), the erosive flux from the bed into suspension is not described at all. The method and equations used to calculate the erosive flux should be added.***

We agree and have rearranged this section and included a new section describing fluxed out of the bed, including the equation for erosive flux. See also our response to referee #2.

***L198-201: It is not instantly clear how the floc size changes in the bed. Deflocculation (L.199) suggests (to me) that flocs degrade to loose sediment particles, but this appears to be in contrast with the preceding statement ("flocs erode as denser, more angular aggregates"). Reading further (and checking the Appendix), I understand that the cohesive size classes tend to an equilibrium distribution, which means that the reverse may also happen: loose clay/silt grains that form aggregates in the bed. Therefore I believe the term "deflocculation" is not well-chosen for this process.***

We agree that "deflocculation" is not the correct term, because the process can go either way. We have changed it to "floc evolution in the bed". However, when larger, less-dense flocs are converted in the bed to smaller, more-dense flocs, they will be available to erode as denser particles...somewhat akin to the observed. We have changed the text in Section 2.2.2 and elsewhere to address this comment, but have not changed the CPP term DEFLOC used in the model code to enable this process.

***L206-221: What happens when the bed is emerged? Are processes like shrinking/swelling accounted for in the bed stratigraphy module, or can these be added in future? Drained clay soils will become more compacted, which is accounted for in the empirical method for the critical bed shear stress. However, are these processes also considered relevant for the determination of the bed layers?***

This is an important question that we have not addressed in the model. We agree that, for accurate representation of intertidal processes, it might be important to account for changes in erodibility by drying (or wetting by rainfall) during low tide. In the current version of the model, layer thickness is related to bulk porosity, but porosity does not change dynamically with compaction...only erodibility is affected. A more process-based model of compaction could be implemented without adding any state

variables, but is not included in this version. We have changed the text in the discussion to list this and several processes that are not included in the model.

**Section 2.4 The method to quantify tau_cb is rather crude. Could the approach be somehow improved by taking the information of the floc size distribution in the bed (Section 2.2) into account? Any reflection and/or suggestions to improve this approach would be useful.**

We agree that the method for setting tau_cb is crude, although we prefer the term "heuristic". A process-based mechanism that relates sediment particle properties (size, density, shape, organic content,…) and measurable geotechnical properties (bulk density, porosity, permeability, shear strength…) would be preferred. However, the approach we have taken can be related to field measurements (e.g., erosion-chamber measurements), so there is some guidance available. The approach is also easily modified when appropriate formulations are accepted in the community.

**L269: The explanations related to P_c are difficult to follow. Insertion of equation S29 from the Supplement would help understanding this section.**

We agree. We have added Eqn. S29 to the Mixed Sediment section as Eqn. 6.

**Section 3: The demonstration cases in Sections 3.1 and 3.2 are very interesting and insightful.**

**L318-325: More explanation regarding the Verney et al. (2011) experiment would be useful. For instance, what is the time of one full cycle in the experiment? Is the dip in the measurements around t = 400 min due to periodicity in velocity forcing, and why doesn't the model reproduce this dip?**

Referee #2 has also commented on this. We have changed the text to clarify the model setup, and to note that the dip in measured grain diameter may have been caused by settling, which was not included in the model simulation.

**L330 introduces the aggregation/collision parameter alfa and break-up/fragmentation parameter beta. Overlooking all test cases in Section 3, alfa varies by a factor 5 and beta by a factor 10. Results appear to be quite sensitive (see e.g. Fig. 3b-c) to the values for alfa and beta. How do values for alfa and beta relate to the physical properties of a cohesive mixture? And how can users determine the optimum value for these parameters? To what extent are the values used for the simulations in Fig.3b accurate (beta <0.02), as they deviate strongly from beta values for the other simulations in the manuscript?**

The values of alpha and beta vary substantially in the different simulations. The rates are adjusted to reproduce the observed (or modeled) data. Ultimately, the magnitudes of alpha and beta are less important than the ratio of alpha/beta, because the ratio defines the relative effectiveness of the competing processes. That ratio does not vary as much between the experiments. More observations are needed to adequately constrain these rates. As of now, user must set the rates in the model to match available floc data. We have not changed the text in response to this comment.

**L430: The active layer is defined as the upper-most layer (L222) which I interpret as being a single grid cell. Consequently I find the explanation in L430 somewhat confusing ("the active layer … extended 2 cm below the surface") given that one grid cell is 1 mm. Can the active layer comprise multiple cells/layers that erode at once, or is the 2cm erosion explained by a stepwise removal of the top "active" layer in 20 time steps?**

The active layer is a single layer at the top. The thickness is determined at each time step according to Harris and Wiberg (1997). If the new thickness increases, material from underlying layers is assimilated; if the new active layer is thinner than it was in the previous time step, it is split into a top, active layer, and an underlying layer. Thickening and thinning of the active layer, in the absence of erosion or deposition, can homogenize the bed down to the depth associated with the thickest active layer. The details of this are described in Section 2.3 of the Supplement, but we have modified the text near L430 to clarify, as follows:

"The first, larger stress event (maximum  = 1 Pa; Figure 5b), eroded 1.2 cm of bed, and expanded the active layer to a thickness of 0.8 cm, so the bed was disturbed to a depth of 2 cm. Expansion of the active layer homogenized enough layers to provide 0.8 cm of sediment, making fine sediment available for resuspension. The finer fractions dominated the suspended sediment in the water column, which contained only a small fraction of the coarsest sand (Figure 5a). When the stress subsided, coarser sediment deposited first, while finer material remained suspended, producing thin layers of graded bedding above the 2-cm limit of initial disturbance (Figure 5d)."

***L460 "compare Figures 6c, d": I understand what I should be seeing, but the differences between the curves are too small to detect them by eye. Perhaps the period with high bed-stress should be extended to make the point.***

We agree that the swelling is imperceptible. Real-world swelling time scales are quite long, so the effect of the swelling is minimal over the simulated time. We plan to run this case for a longer period and modify the figure to clarify this.

***Fig. 3a: what do the error bars depict? 95% C-I, or +/- 1*st.dev?***

Per text in Verney et al (2011), these represent +/- one standard deviation about the mean diameter D. We have modified the caption for Figure 3 to note this.

***Technical corrections***

***L78-79: "that that"***

Fixed.

***L104: "seagrass growth model" ⌦ models?***

Corrected.

***L335 full stop missing at end of sentence.***

Added.

***L513 last sentence refers to Figure 8a, but no information on the grain size is contained in this figure. Consequently also the title of Fig. 8a is incorrect.***

We agree this is confusing. This is referring to the simulation without floc dynamics, in which all of the sediment in suspension is in the 37-um size class. The text has been changed to read: "No floc dynamics were included, so all of the suspended material depicted in Figure 8a was in the 37-$\mu$m class."

The caption to Figure 8 has been changed to read: "Figure 8. Comparison of estuarine turbidity maxima simulations with and without floc dynamics. a) Two-dimensional (along-estuary and vertical) snapshot of suspended particle concentrations (shaded) without floc dynamics near the end of flood tide. All of the suspended material was in the 37-μm class. b) Snapshot of suspended particle concentrations at the same time in the simulation, but with simulated floc dynamics (shading), overlain by contours of mean particle diameters. c) Along-estuary profiles of bed elevations for simulations without floc dynamics (red) and with floc dynamics (black) at the peak of flood tide (solid lines) and at post-flood slack tide (dashed lines). d) Along-estuary profiles of mean particle diameter in the top layer of the seabed, using the same notation as (c)."

---

## Short Comment (SC3) · 15 Mar 2018

Response to interactive comment by Anonymous Referee #2 on "Cohesive and mixed sediment in the Regional Ocean Modeling System (ROMS v3.6) implemented in the Coupled Ocean Atmosphere Wave Sediment-Transport Modeling System (COAWST r1179)" by Christopher R. Sherwood et al. Comment received 24 January 2018.

The authors thank Anonymous Referee #2 for thoughtful and helpful comments on our manuscript. Here, we respond to those comments and indicate changes we have made in the manuscript to address them. Comments are reproduced in **bold+italics**; our response is in plain text.

***The authors present the implementation of a cohesive and mixed sediment module within the COAWST (ROMS based system). They provide a thorough and extensive framework that includes floc model, stratigraphy and bed mixing, critical stress for erosion of cohesive sediment. None of the individual components is particularly novel in isolation, but the overall model combining all aspects does present a significant advance in coastal sediment transport modelling. The manuscript is well written and I enjoyed reading it.***

Thank you for these complimentary words. We agree that none of the components is novel in isolation, but hope that we have constructed a useful modeling framework.

***There are a few issues that would need to be addressed in a revision.***

***The most important issue is that it is not clear how the floc model is combined with the vertical ROMS grid and vertical sediment fluxes (turbulent suspension and settling) to determine suspensions of cohesive sediments. Are these actually included (the steady state test suggests yes but the comparison to Verney (2011) no)? The key discrepancy in the model-data comparison in figure 3a at t=400 min corresponds to a settling stage. In Verney et al. (2011), the settling dip was not reproduced either as particle deposition was not allowed in the 0D model. Is the same explanation also valid here?***

The floc model is a zero-dimensional model that is locally integrated over the baroclinic time step, from initial to final conditions, in every cell of the ROMS model. After the floc populations are updated, the normal settling, advection, and diffusion routines in ROMS are advanced, with flux boundary conditions at the bed (erosion or deposition) and zero-flux conditions at the surface. This transport generally changes the floc populations in model cells, providing new initial population conditions for the next time step.

The steady-state test (Fig. 4) is a fully three-dimensional implementation, but the horizontal aspect of the grid is small (5 cells…just enough to accommodate the templates of the finite-difference formulations) and lateral periodic boundary conditions are applied, so that anything advected out of the domain re-enters on the upstream side. Therefore, it is effectively a one-dimensional (vertical) simulation. To reproduce the results of Verney et al. (2011), we set the settling velocities of all floc classes to zero and imposed the turbulent shear parameter, so that the simulation is effectively zero-dimensional with constant suspended mass, and the only active process in the model is the floc dynamics. Thus, our implementation has the same shortcomings as the Verney (2011) implementation, in that we cannot assess changes that might be caused by settling.

Simulations with advection, diffusion and settling are included in the other experiments of the paper.

We have added text in Sections 2.2, 3.1.1, and 3.1.3 for clarification.

*Another weakness is that, even though the manuscript includes a number of test cases, it looks to me that there is a lack of validation. Only the floc model is validated against measurements and there is no validation against field observations, especially for cohesive suspended sediments. This is somewhat frustrating and looks like a missed opportunity as LISST instruments are now relatively commonly deployed in the field. Since they measure concentrations for a number of floc size classes, they would appear to be well suited to provide datasets for validation and model-observation comparisons.*

Validation of each component of the model would substantially expand the scope of an already lengthy paper. Comparisons of each component of the model with field data would require introduction of the observations and analysis of the inevitable discrepancies between the model and data. We have collected a LISST dataset similar to that suggested by the Referee (Sherwood et al., 2012. USGS Open-File Report 1178, https://pubs.usgs.gov/of/2012/1178/title_page.html) and we plan to compare it against the model. The final section of our manuscript provides some comparison of the cohesive bed component with real-world observations. Otherwise, we hope that our demonstrations that model components work and produce plausible results provides sufficient guidance and incentive for others to apply and evaluate the model. The goal of the paper is methodological and we demonstrate the potential applications of the newly implemented routines. We have not changed the manuscript to address this comment.

*Given that the new algorithms are incorporated in COAWST, I am wondering about coupling and/or compatibility with the wave module(s). While a full test of this may be outside of the scope of the paper, I think discussing this point would strengthen the manuscript.*

The Referee is correct in noting that waves are closely coupled in the COAWST system, which allows two-way coupling between ROMS and either WaveWatch III or SWAN. Within ROMS, waves have several effects: a) wave-induced momentum fluxes (implemented as either vortex forcing or radiation stresses) drive circulation; b) wave breaking affects near-surface turbulence; and c) wave- and current-combined bottom stresses affect sediment resuspension and near-bed turbulence. All of these, especially the last, have direct implications for cohesive sediment processes. However, ROMS does not have a stress-strain relationship suitable for simulating the visco-elastic behavior of very high concentrations of mud. We have added text at the beginning of the Discussion to address this comment, as follows: "The improvements were implemented in the COAWST version of ROMS, which provides a framework for realistic two-way nested models with forcing from meteorology (WRF; Michalakes et al., 2001) and waves (either SWAN: Booj et al., 1999; or WaveWatch III; Tolman et al., 2014). Waves, in particular, play an important role in cohesive sediment dynamics through wave-enhanced bottom shear stresses, wave-induced near-bottom turbulence, and wave-induced nearshore circulation, but wave-induced fluid-mud layer processes are not represented."

*Specific comments:*

**Section 2.2.1: I'm not sure whether this is the best place to present fluxes into the bed. The alternative (which probably would be my preference) is to combine with erosion into a "bed water column exchange" section.**

This is a good suggestion, and we have re-arranged the paper to address it. We have added heading "2.3 Bed – Water Column Exchange" with subsection "2.3.1. Fluxes into the bed – Critical shear stress for deposition" (with the contents of previous Section 2.2.1) and a new subsection "2.3.2. Fluxes out of the bed – Resuspension" which includes the erosion rate equation.

Previous section "2.3.2. Changes in floc size distribution within the bed" has been moved up as Section 2.2.3. We thank the reviewer for helping make this section clearer and more readable.

**Figure 3a,b: It would be helpful to also have the temporal evolution of G shown. Since the authors include the modelling results of Verney et al. (2011), it would be useful to explain the reason for the different model results during the first aggregation stage (initial distribution), instead of relying on the (initial distribution), instead of relying on the reader checking in Verney et al. (2011).**

We agree. We have added time-dependent curves for G to Fig 3 a in the revised manuscript.

**Section 2.6: The new modules are added to the existing sediment transport model in ROMS (Warner et al., 2008) and in COAWST, which includes waves. The presence of bedforms and waves may induce pressure gradients at the sediment bed, which would in turn induce interstitial porewater flow in the bed. This process can entrain fine particles into a coarser sediment bed (e.g., Huettel et al., 1996, Limnol. Oceanogr.,41(2), 1996, 309-322). It would be welcome for the authors to comment on this process and its inclusion (or not) in the present framework.**

We agree this process might be important, especially for biogeochemical constituents. It is not represented in this version of ROMS because small scale bottom topography is not resolved and our version of ROMS is not yet coupled with a groundwater transport model, and we have not explored a sub-grid scale parameterization of this process. We have added to the discussion a short list of processes that are not addressed in the model, as follows: "However, not all of the processes associated with cohesive or mixed sediment have been included. For example, fluid muds and non-Newtonian flows are not represented (e.g., Mehta, 1991; 2014), nor is flow-induced infiltration of fine material into a porous bed (Huettel et al., 1999). Changes to the erodibility of mud that has been exposed at low tide (e.g., Paterson et al, 1990; Pilditch et al., 2008) or affected by flora or fauna (e.g., de Boer, 1981; de Deckere et al., 2001) are not considered."

**Figure 4: there appears to be a "kink" in the concentration for one specific profile (3560 microns?). What is the cause?**

The model solution becomes very sensitive, especially for larger particles, when both C and G are high, so it produces instabilities. We are not sure if these are real, or numerical artifacts, but they only occur under conditions with very high concentrations and turbulent shear.

**Figure 8: Caption should include details on what the different panels (a, b, c, d) show.**

We agree, and are not sure where those details went! We changed the caption to read as follows: "Figure 8. Comparison of estuarine turbidity maxima simulations with and without floc dynamics. a) Two-dimensional (along-estuary and vertical) snapshot of suspended particle concentrations (shaded)

without floc dynamics near the end of flood tide. b) Snapshot of suspended particle concentrations at the same time in the simulation, but with simulated floc dynamics (shading), overlain by contours of mean particle diameters. c) Along-estuary profiles of bed elevations for simulations without floc dynamics (red) and with floc dynamics (black) at the peak of flood tide (solid lines) and at post-flood slack tide (dashed lines). d) Along-estuary profiles of mean particle diameter in the top layer of the seabed, using the same notation as (c). The model was initialized with a uniform suspended-sediment concentration of 0.1 kg/m3 in the 37-μm class."

***Technical corrections:***

***Line 79: one too many that***

Fixed.

***Line 145-146 vs lines 115-116: Repetition, please remove one of the two.***

We modified the text near lines 145-146 to help address the Referees first comment, so there is no longer repetition.

---

## Referee Comment (RC3) · Anonymous Referee #1 · 17 Mar 2018

The authors have given a thoughtful response and improved the manuscript according to my suggestions. In my opinion the manuscript is ready for publication. I wish to congratulate the authors and thank them for their efforts.
* * *

---

## Author Comment (AC1) · 19 Mar 2018

Thanks for referee #1 for his/her supportive response to our response.

This note is a reminder that the official authors' response to both refereee # 1 and # 2 are contained in the .pdf attachments to the short notes posted by co-author Alfredo Aretxabaleta.

[Figure]

-Chris Sherwood

---

## Referee Comment (RC4) · Anonymous Referee #3 · 20 Mar 2018

Manuscript Number: GMD-2017-267-V1 Full Title: Cohesive and mixed sediment in the Regional Ocean Modeling System (ROMS) Article Type: Research Paper Authors: Christopher R. Sherwood, Alfredo L. Aretxabaleta, Courtney K. Harris, J. Paul Rinehimer, Romaric Verney, Bénédicte Ferré

OVERVIEW OF THE MS: This manuscript describes and demonstrates algorithms for treating fine and cohesive sediment that have been implemented in the Regional Ocean

[Figure]

Modeling System (ROMS). These include: floc dynamics (aggregation and disaggregation in the water column); changes in floc characteristics in the seabed; erosion and deposition of cohesive and mixed (combination of cohesive and non-cohesive) sediment; and biodiffusive mixing of bed sediment. These routines supplement existing non-cohesive sediment modules, thereby increasing our ability to model fine-grained and mixed-sediment environments. Additionally, the manuscript describes changes to the sediment bed-layering scheme that improve the fidelity of the modeled stratigraphic record. Finally, the manuscript provides examples of these modules implemented in idealized test cases and a realistic application.

———————————————————————————————————————————— MY

REVIEW COMMENTS:

I see these finding to be very interesting and of great importance, especially for coastal environmental management, where the accurate prediction of the movement and transport of both purely cohesive and mixed sediments is vital, for issues such as navigational waterways and water quality.

The manuscript is generally well written and correctly structured, some relevant illustrations, and an appropriate range of relevant literature cited and referenced. The study aims and objectives are clearly defined on pp 4.

However, the following points need to be addressed in detail, before this manuscript can be considered for publication.

Well written abstract. I would like to see a few more key quantitative findings reported there, in particular in terms of typical SSC levels and hydrodynamic ranges assessed by the model, plus some key model output values. I would also suggest doing the same for the Conclusion (pp30-31).

In Section 2 – Model Processes: I would like to see a little more background on sediment transport process theory. This would assist the reader with fundamentals behind

how the new model opporates.

In Section 2.2 – Floc Processes: again, I think this section would benefit by having some brief flocculation theory review presented before the floc model description.

I think it would be good to briefly outline the range of different approaches used in flocculation modeling, and why the approach used in this model was chosen.

Other aspects that I would like to see further updated in the manuscript, are slight updates with the Introduction section, where specific aspects could be further strengthened. I would like to recommend including some of the following references in the Introduction literature review. This would significantly strengthen the literature reviewed in the manuscript. These would provide links to recent research findings that would provide synergy and context for the research reported in this manuscript. It would be good if aspects of the following publications were included in the Discussion.

These four publications provide additional insights into cohesive sediment flocculation and associated settling dynamics, together with applied modelling: - Mehta, A.J., Manning, A.J. and Khare, Y.P. (2014). A Note on the Krone deposition equation and significance of floc aggregation. Marine Geology, 354, 34-39, doi.org/10.1016/j.margeo.2014.04.002. - Mietta, F., Chassagne, C., Manning, A.J. and Winterwerp, J.C. (2009). Influence of shear rate, organic matter content, pH and salinity on mud flocculation. Ocean Dynamics, 59, 751-763, doi: 10.1007/s10236-009-0231-4. - Soulsby, R.L., Manning, A.J., Spearman, J. and Whitehouse, R.J.S. (2013). Settling velocity and mass settling flux of flocculated estuarine sediments. Marine Geology, doi.org/10.1016/j.margeo.2013.04.006. - Winterwerp, J.C., Manning, A.J., Martens, C., de Mulder, T., and Vanlede, J. (2006). A heuristic formula for turbulence-induced flocculation of cohesive sediment. Estuarine, Coastal and Shelf Science, 68, 195-207.

These two publications have demonstrated the importance of biological cohesion on bed sediments, as this has an important role on erosion threshold and bio-stability: -
Malarkey, J., Baas, J.H., Hope, J.A., Aspden, R.J., Parsons, D.R., Peakall, J., Paterson, D.M., Schindler, R.J., Ye, L., Lichtman, I.D., Bass, S.J., Davies, A.G., Manning, A.J., Thorne, P.D. (2015). The pervasive role of biological cohesion in bedform development. Nature Communications, DOI: 10.1038/ncomms7257. - Parsons, D.R., Schindler, R.J., Hope, J.A., Malarkey, J., Baas, J.H., Peakall, J., Manning, A.J., Ye, L., Simmons, S., Paterson, D.M., Aspden, R.J., Bass, S.J., Davies, A.G., Lichtman, I.D. and Thorne, P.D. (2016). The role of biophysical cohesion on subaqueous bed form size. Geophysical Research Letters, 43, doi:10.1002/2016GL067667.

This publication provides good general overviews of cohesive sediment dynamics: - Mehta, A.J. (2014). An Introduction to Hydraulics of Fine Sediment Transport, World Scientific, Hackensack, N. J.

Although the manuscript mentions mixed sediments in Section 2.5, it reports very little about the effects of mixed sediment flocculation. As much of the model application could be utilized in areas where there are sand / silt / clay, and biological cohesions, the manuscript would benefit from the citation of some of these recent key publications on the flocculation processes of cohesive and mixed fine-grained sediment suspension, as these outline key processes relating to these suspended sediment types:

* Manning, A.J., Baugh, J.V., Spearman, J.R., Pidduck, E.L. and Whitehouse, R.J.S. (2011). The settling dynamics of flocculating mud:sand mixtures: Part 1 – Empirical algorithm development. Ocean Dynamics, INTERCOH 2009 special issue, doi: 10.1007/s10236-011-0394-7. * Manning, A.J., Baugh, J.V., Spearman, J. and Whitehouse, R.J.S. (2010). Flocculation Settling Characteristics of Mud:Sand Mixtures. Ocean Dynamics, doi: 10.1007/s10236-009-0251-0. * Spearman, J.R., Manning, A.J. and Whitehouse, R.J.S. (2011). The settling dynamics of flocculating mud:sand mixtures: Part 2 – Numerical modelling. Ocean Dynamics, doi: 10.1007/s10236-011-0385-8.

In terms of the erosion-depositional cycle, Spearman and Manning (2008) have

demonstrated that the threshold shear stresses for both deposition and erosion can operate simultaneously, in order to correctly mass-balance accretion and erosion levels of cohesive sediments during tidal cycles in shallow water locations. I would like to see this commented on within the context of your own study findings. - Spearman, J. and Manning, A.J. (2008). On the significance of mud transport algorithms for the modelling of intertidal flats. In: T. Kudusa, H. Yamanishi, J. Spearman and J.Z. Gailani, (Eds.), Sediment and Ecohydraulics - Proc. in Marine Science 9, Amsterdam: Elsevier, pp. 411-430, ISBN: 978-0-444-53184-1.

I would like to see the Discussion (Section 5) expanded slightly, with some comparisons made with other commonly used sediment transport modeling approaches. Some quatification (also in a summary Table) to these comparisons would be helpful. This could advise the reader on where significant improvements and advances have been made with this new modeling approach. It would also be good to comment on the possible limitations on this new modeling approach.

In summary, I think these findings are significant and are worthy of publication in GMD.

---

## Author Response (AR1)

Final author response to interactive comments by three Anonymous Referees and the Editor on "Cohesive and mixed sediment in the Regional Ocean Modeling System (ROMS v3.6) implemented in the Coupled Ocean Atmosphere Wave Sediment-Transport Modeling System (COAWST r1179)" by Christopher R. Sherwood et al.

We thank the three anonymous referees for helpful and constructive comments on our draft manuscript. We especially thank editor Guy Munhoven for enlisting the referees and helping to moderate the discussion.

The comments have led us to make moderate but important revisions to the manuscript. We have added 23 references and modified the table and three of nine figures. We have rearranged section 2 (Model Processes) and added an equation to clarify our presentation of model processes. Most importantly, we have expanded our introduction and discussion to address issues raised by the referees. We hope these changes make our presentation clearer and more compelling.

The title of the final paper will be changed to reflect the most recent source code repository revision number.

Response to interactive comment by Anonymous Referee #1 on "Cohesive and mixed sediment in the Regional Ocean Modeling System (ROMS v3.6) implemented in the Coupled Ocean Atmosphere Wave Sediment-Transport Modeling System (COAWST r1179)" by Christopher R. Sherwood et al. Comment received 20 December, 2017.

The authors thank Anonymous Referee #1 for detailed and insightful comments on our manuscript. Here, we respond to those comments and indicate changes we have made in the manuscript to address them. Comments are reproduced in **bold+italics**; our response is in plain text.

*1. Does the paper address relevant scientific modelling questions within the scope of GMD? Does the paper present a model, advances in modelling science, or a modelling protocol that is suitable for addressing relevant scientific questions within the scope of EGU?*

*The authors extended an existing model for regional-scale coastal sediment transport and morphodynamics by implementing a number of previously developed routines that account for cohesive sediment and biogemorphology effects. The upgraded model is most likely of interest to both academics and engineers working in the coastal community.*

*2. Does the paper present novel concepts, ideas, tools, or data?*

*The present study does not present completely new model concepts, but instead, it combines existing model formulations that were developed by the same authors in preceding studies (Warner et al., 2008; Rinehimer et al., 2008; Verney et al., 2011). This leads to an upgraded version of the ROMS model, which is considered a novel tool that is worthy of publication.*

*3. Does the paper represent a sufficiently substantial advance in modelling science?*

*Yes.*

*4. Are the methods and assumptions valid and clearly outlined?*

*The implemented methods have been described in preceding studies and seem valid. However, some components in the model and underlying assumptions require additional clarification, see my specific remarks below.*

*5. Are the results sufficient to support the interpretations and conclusions?*

*The authors present results of a number of idealized "demonstration cases" and a realistic application. These cases are generally interesting and the results support the conclusions. Specific remarks regarding the simulations and the interpretations of results are listed below.*

*6. Is the description sufficiently complete and precise to allow their reproduction by fellow scientists (traceability of results)? In the case of model description papers, it should in theory be possible for an independent scientist to construct a model that, while not necessarily numerically identical, will produce scientifically equivalent results. Model development papers should be similarly reproducible. For MIP and benchmarking papers, it should be possible for the protocol to be precisely reproduced for an independent model. Descriptions of numerical advances should be precisely reproducible.*

*The explanations are at some points rather short, and for a full understanding of the methodology (e.g. equations and numerical implementation) the reader has to turn to preceding papers by these authors and to information contained in the Supplement. I appreciate, however, that a journal format may not allow to fully explain all the details. Given that the numeric code is available to anyone, and that the community is explicitly invited to use the code, I expect the present work to be fully reproducible.*

*7. Do the authors give proper credit to related work and clearly indicate their own new/original contribution?*

*Yes.*

*8. Does the title clearly reflect the contents of the paper? The model name and number should be included in papers that deal with only one model.*

*Yes.*

*9. Does the abstract provide a concise and complete summary?*

*Yes.*

*10. Is the overall presentation well structured and clear?*

*Yes.*

*11. Is the language fluent and precise?*

*The paper is well-written in fluent English.*

*12. Are mathematical formulae, symbols, abbreviations, and units correctly defined and used?*

*Yes.*

*13. Should any parts of the paper (text, formulae, figures, tables) be clarified, reduced, combined, or eliminated?*

*Yes, see specific comments below.*

*14. Are the number and quality of references appropriate?*

*Yes.*

*15. Is the amount and quality of supplementary material appropriate? For model description papers, authors are strongly encouraged to submit supplementary material containing the model code and a user manual. For development, technical, and benchmarking papers, the submission of code to perform calculations described in the text is strongly encouraged.*

*The 27-page Supplement provides details on the implemented methodology, including a description of the main equations. The code is not directly provided but is available upon request.*

Thank you for this comprehensive review.

*Specific major comments*

*1. Given that one of the model goals is to simulate morphologic change (Line 98), I am surprised that the realistic application of the model to the York River Estuary does not address the morphologic evolution at all. Is the model also capable of accurately simulating longer-term morphologic changes in a complex environment such as an estuary? If the authors were to run the model for a longer simulation time (say a few years), would the model reproduce a reliable evolution of the main geomorphologic features (banks, creeks, shoals, …) of the estuary? To me this is a key issue in trusting the model's performance, and results or a general discussion on this issue are essential.*

*It would also be interesting to see how the modeled morphology would differ for simulations with the present, upgraded model, relative to simulations with the original model by Warner et al. (2008).*

We agree that validation of the model for long simulations of geomorphological evolution is needed. And we admit that it will be a challenge to match observations of geomorphological change in cohesive environments, and can't affirm that the model will reliably reproduce changes in banks, creeks, or shoals. But, as we responded to Referee #2, validation of each component of the model would substantially expand the scope of an already lengthy paper. Comparisons of each component of the model with field data would require introduction of the observations and analysis of the inevitable discrepancies between the model and data. The goal of the paper is to describe the modeling methods, and we hope that our demonstrations of potential applications, which produce plausible results, provide sufficient guidance and incentive for others to apply and evaluate the model. We look forward to doing so ourselves. We have not changed the manuscript to address this comment.

*2. A topic that is overlooked, or at least not considered in the manuscript, is bedload transport - apart from a general notion that the stratigraphy is relevant for bedload transport (L.207). This is rather confusing and I believe the following topics should be addressed:*

*A. Is size-selective bedload transport included at all? If yes, which model is used?*

*B. How does the bedload transport depend on the particle size distribution in e.g. the active layer?*

*C. How is the critical bed shear stress for bedload determined? Is the applied method consistent with the methodology proposed for the erosion rate in Section 2.4?*

A. Yes. The CSTMS bedload transport equations included in ROMS (Warner et al., 2008) are available and suitable for transporting the non-cohesive components in a mixed bed simulation. There are presently two options: the Meyer-Peter Mueller equation, or the Soulsby equations that include asymmetric transport by waves. The transport rates are size dependent, as discussed below.

B. These equations use the user-specified particle critical shear stress for erosion for each size class, and act on any non-cohesive classes present in the top (active) layer when Tb exceeds Tcrit for that size class AND Tau_b > Tau_cb when mixed sediment is present. In other words, a sand grain embedded in a cohesive bed will not move unless the bed stress is both greater than the bulk critical shear stress of the bed and the particle critical shear stress needed to mobilize the sand grain. We assume that cohesive sediment does not undergo bedload transport; eroded cohesive material goes directly into suspension.

C. The critical bed shear stress for bedload in a mixed bed is the critical particle shear stress computed from, for example, a Shields relationship. However, the material will not undergo bedload transport unless the bulk critical shear stress for the bed (as described in Section 2.5 [now 2.6]) is exceeded. The erosion rate (flux from the bed into suspension) is governed by the greater of the two critical shear stress values. We don't think there is inconsistency in this approach, but it does assume that the presence of cohesive sediment does not affect the bedload transport rates of available non-cohesive sediment.

Text has been added to the end of Section 2.5 [now 2.6] as follows:

"Non-cohesive sediment classes are subject to bedload transport when the bottom stress exceeds both the bulk critical shear stress of the top (active) layer and the particle critical shear stress for that class. In these cases, the transport-rate equations still calculate bedload transport based on excess shear stress associated with the non-cohesive particle critical shear stress, as described in Warner et al (2008). Cohesive classes are not subject to bedload transport; if the bulk critical shear stress of the bed is exceeded, we assume they will go directly into suspension."

We thank the reviewer for bringing up this issue, because it led us to an error in the code that will be fixed in the release accompanying the final manuscript.

***Minor comments***

***Section 2: While a section is devoted to the flux into the bed (2.2.1), the erosive flux from the bed into suspension is not described at all. The method and equations used to calculate the erosive flux should be added.***

We agree and have rearranged this section and included a new section describing fluxed out of the bed, including the equation for erosive flux. See also our response to referee #2.

***L198-201: It is not instantly clear how the floc size changes in the bed. Deflocculation (L.199) suggests (to me) that flocs degrade to loose sediment particles, but this appears to be in contrast with the preceding statement ("flocs erode as denser, more angular aggregates"). Reading further (and checking the Appendix), I understand that the cohesive size classes tend to an equilibrium distribution, which means that the reverse may also happen: loose clay/silt grains that form aggregates in the bed. Therefore I believe the term "deflocculation" is not well-chosen for this process.***

We agree that "deflocculation" is not the correct term, because the process can go either way. We have changed it to "floc evolution in the bed". However, when larger, less-dense flocs are converted in the bed to smaller, more-dense flocs, they will be available to erode as denser particles…somewhat akin to the observed. We have changed the text in Section 2.2.2 and elsewhere to address this comment, but have not changed the CPP term DEFLOC used in the model code to enable this process.

***L206-221: What happens when the bed is emerged? Are processes like shrinking/swelling accounted for in the bed stratigraphy module, or can these be added in future? Drained clay soils will become more compacted, which is accounted for in the empirical method for the critical bed shear stress. However, are these processes also considered relevant for the determination of the bed layers?***

This is an important question that we have not addressed in the model. We agree that, for accurate representation of intertidal processes, it might be important to account for changes in erodibility by drying (or wetting by rainfall) during low tide. In the current version of the model, layer thickness is related to bulk porosity, but porosity does not change dynamically with compaction…only erodibility is affected. A more process-based model of compaction could be implemented without adding any state variables, but is not included in this version. We have changed the text in the discussion to list this and several processes that are not included in the model.

**Section 2.4 The method to quantify tau_cb is rather crude. Could the approach be somehow improved by taking the information of the floc size distribution in the bed (Section 2.2) into account? Any reflection and/or suggestions to improve this approach would be useful.**

We agree that the method for setting tau_cb is crude, although we prefer the term "heuristic". A process-based mechanism that relates sediment particle properties (size, density, shape, organic content,…) and measurable geotechnical properties (bulk density, porosity, permeability, shear strength…) would be preferred. However, the approach we have taken can be related to field measurements (e.g., erosion-chamber measurements), so there is some guidance available. The approach is also easily modified when appropriate formulations are accepted in the community.

**L269: The explanations related to P_c are difficult to follow. Insertion of equation S29 from the Supplement would help understanding this section.**

We agree. We have added Eqn. S29 to the Mixed Sediment section as Eqn. 6.

**Section 3: The demonstration cases in Sections 3.1 and 3.2 are very interesting and insightful.**

**L318-325: More explanation regarding the Verney et al. (2011) experiment would be useful. For instance, what is the time of one full cycle in the experiment? Is the dip in the measurements around t = 400 min due to periodicity in velocity forcing, and why doesn't the model reproduce this dip?**

Referee #2 has also commented on this. We have changed the text to clarify the model setup, and to note that the dip in measured grain diameter may have been caused by settling, which was not included in the model simulation.

**L330 introduces the aggregation/collision parameter alfa and break-up/fragmentation parameter beta. Overlooking all test cases in Section 3, alfa varies by a factor 5 and beta by a factor 10. Results appear to be quite sensitive (see e.g. Fig. 3b-c) to the values for alfa and beta. How do values for alfa and beta relate to the physical properties of a cohesive mixture? And how can users determine the optimum value for these parameters? To what extent are the values used for the simulations in Fig.3b accurate (beta <0.02), as they deviate strongly from beta values for the other simulations in the manuscript?**

The values of alpha and beta vary substantially in the different simulations. The rates are adjusted to reproduce the observed (or modeled) data. Ultimately, the magnitudes of alpha and beta are less important than the ratio of alpha/beta, because the ratio defines the relative effectiveness of the competing processes. That ratio does not vary as much between the experiments. More observations are needed to adequately constrain these rates. As of now, user must set the rates in the model to match available floc data. We have not changed the text in response to this comment.

**L430: The active layer is defined as the upper-most layer (L222) which I interpret as being a single grid cell. Consequently I find the explanation in L430 somewhat confusing ("the active layer … extended 2 cm below the surface") given that one grid cell is 1 mm. Can the active layer comprise multiple cells/layers that erode at once, or is the 2cm erosion explained by a stepwise removal of the top "active" layer in 20 time steps?**

The active layer is a single layer at the top. The thickness is determined at each time step according to Harris and Wiberg (1997). If the new thickness increases, material from underlying layers is assimilated; if the new active layer is thinner than it was in the previous time step, it is split into a top, active layer, and an underlying layer. Thickening and thinning of the active layer, in the absence of erosion or deposition, can homogenize the bed down to the depth associated with the thickest active layer. The details of this are described in Section 2.3 of the Supplement, but we have modified the text near L430 to clarify, as follows:

"The first, larger stress event (maximum = 1 Pa; Figure 5b), eroded 1.2 cm of bed, and expanded the active layer to a thickness of 0.8 cm, so the bed was disturbed to a depth of 2 cm. Expansion of the active layer homogenized enough layers to provide 0.8 cm of sediment, making fine sediment available for resuspension. The finer fractions dominated the suspended sediment in the water column, which contained only a small fraction of the coarsest sand (Figure 5a). When the stress subsided, coarser sediment deposited first, while finer material remained suspended, producing thin layers of graded bedding above the 2-cm limit of initial disturbance (Figure 5d)."

***L460 "compare Figures 6c, d": I understand what I should be seeing, but the differences between the curves are too small to detect them by eye. Perhaps the period with high bed-stress should be extended to make the point.***

We agree that the swelling is imperceptible. Real-world swelling time scales are quite long, so the effect of the swelling is minimal over the simulated time. We plan to run this case for a longer period and modify the figure to clarify this.

***Fig. 3a: what do the error bars depict? 95% C-I, or +/- 1*st.dev?***

Per text in Verney et al (2011), these represent +/- one standard deviation about the mean diameter D. We have modified the caption for Figure 3 to note this.

***Technical corrections***

***L78-79: "that that"***

Fixed.

***L104: "seagrass growth model" ⬜ models?***

Corrected.

***L335 full stop missing at end of sentence.***

Added.

***L513 last sentence refers to Figure 8a, but no information on the grain size is contained in this figure. Consequently also the title of Fig. 8a is incorrect.***

We agree this is confusing. This is referring to the simulation without floc dynamics, in which all of the sediment in suspension is in the 37-um size class. The text has been changed to read: "No floc dynamics were included, so all of the suspended material depicted in Figure 8a was in the 37-$\mu$m class."

The caption to Figure 8 has been changed to read: "Figure 8. Comparison of estuarine turbidity maxima simulations with and without floc dynamics. a) Two-dimensional (along-estuary and vertical) snapshot of suspended particle concentrations (shaded) without floc dynamics near the end of flood tide. All of the suspended material was in the 37-μm class. b) Snapshot of suspended particle concentrations at the same time in the simulation, but with simulated floc dynamics (shading), overlain by contours of mean particle diameters. c) Along-estuary profiles of bed elevations for simulations without floc dynamics (red) and with floc dynamics (black) at the peak of flood tide (solid lines) and at post-flood slack tide (dashed lines). d) Along-estuary profiles of mean particle diameter in the top layer of the seabed, using the same notation as (c)."

Response to interactive comment by Anonymous Referee #2 on "Cohesive and mixed sediment in the Regional Ocean Modeling System (ROMS v3.6) implemented in the Coupled Ocean Atmosphere Wave Sediment-Transport Modeling System (COAWST r1179)" by Christopher R. Sherwood et al. Comment received 24 January 2018.

The authors thank Anonymous Referee #2 for thoughtful and helpful comments on our manuscript. Here, we respond to those comments and indicate changes we have made in the manuscript to address them. Comments are reproduced in **_bold+italics_**; our response is in plain text.

**_The authors present the implementation of a cohesive and mixed sediment module within the COAWST (ROMS based system). They provide a thorough and extensive framework that includes floc model, stratigraphy and bed mixing, critical stress for erosion of cohesive sediment. None of the individual components is particularly novel in isolation, but the overall model combining all aspects does present a significant advance in coastal sediment transport modelling. The manuscript is well written and I enjoyed reading it_**.

Thank you for these complimentary words. We agree that none of the components is novel in isolation, but hope that we have constructed a useful modeling framework.

**_There are a few issues that would need to be addressed in a revision._**

**_The most important issue is that it is not clear how the floc model is combined with the vertical ROMS grid and vertical sediment fluxes (turbulent suspension and settling) to determine suspensions of cohesive sediments. Are these actually included (the steady state test suggests yes but the comparison to Verney (2011) no)? The key discrepancy in the model-data comparison in figure 3a at t=400 min corresponds to a settling stage. In Verney et al. (2011), the settling dip was not reproduced either as particle deposition was not allowed in the 0D model. Is the same explanation also valid here?_**

The floc model is a zero-dimensional model that is locally integrated over the baroclinic time step, from initial to final conditions, in every cell of the ROMS model. After the floc populations are updated, the normal settling, advection, and diffusion routines in ROMS are advanced, with flux boundary conditions at the bed (erosion or deposition) and zero-flux conditions at the surface. This transport generally changes the floc populations in model cells, providing new initial population conditions for the next time step.

The steady-state test (Fig. 4) is a fully three-dimensional implementation, but the horizontal aspect of the grid is small (5 cells...just enough to accommodate the templates of the finite-difference formulations) and lateral periodic boundary conditions are applied, so that anything advected out of the domain re-enters on the upstream side. Therefore, it is effectively a one-dimensional (vertical) simulation. To reproduce the results of Verney et al. (2011), we set the settling velocities of all floc classes to zero and imposed the turbulent shear parameter, so that the simulation is effectively zero-dimensional with constant suspended mass, and the only active process in the model is the floc dynamics. Thus, our implementation has the same shortcomings as the Verney (2011) implementation, in that we cannot assess changes that might be caused by settling.

Simulations with advection, diffusion and settling are included in the other experiments of the paper.

We have added text in Sections 2.2, 3.1.1, and 3.1.3 for clarification.

*Another weakness is that, even though the manuscript includes a number of test cases, it looks to me that there is a lack of validation. Only the floc model is validated against measurements and there is no validation against field observations, especially for cohesive suspended sediments. This is somewhat frustrating and looks like a missed opportunity as LISST instruments are now relatively commonly deployed in the field. Since they measure concentrations for a number of floc size classes, they would appear to be well suited to provide datasets for validation and model-observation comparisons.*

Validation of each component of the model would substantially expand the scope of an already lengthy paper. Comparisons of each component of the model with field data would require introduction of the observations and analysis of the inevitable discrepancies between the model and data. We have collected a LISST dataset similar to that suggested by the Referee (Sherwood et al., 2012. USGS Open-File Report 1178, https://pubs.usgs.gov/of/2012/1178/title_page.html) and we plan to compare it against the model. The final section of our manuscript provides some comparison of the cohesive bed component with real-world observations. Otherwise, we hope that our demonstrations that model components work and produce plausible results provides sufficient guidance and incentive for others to apply and evaluate the model. The goal of the paper is methodological and we demonstrate the potential applications of the newly implemented routines. We have not changed the manuscript to address this comment.

*Given that the new algorithms are incorporated in COAWST, I am wondering about coupling and/or compatibility with the wave module(s). While a full test of this may be outside of the scope of the paper, I think discussing this point would strengthen the manuscript.*

The Referee is correct in noting that waves are closely coupled in the COAWST system, which allows two-way coupling between ROMS and either WaveWatch III or SWAN. Within ROMS, waves have several effects: a) wave-induced momentum fluxes (implemented as either vortex forcing or radiation stresses) drive circulation; b) wave breaking affects near-surface turbulence; and c) wave- and current-combined bottom stresses affect sediment resuspension and near-bed turbulence. All of these, especially the last, have direct implications for cohesive sediment processes. However, ROMS does not have a stress-strain relationship suitable for simulating the visco-elastic behavior of very high concentrations of mud. We have added text at the beginning of the Discussion to address this comment, as follows: "The improvements were implemented in the COAWST version of ROMS, which provides a framework for realistic two-way nested models with forcing from meteorology (WRF; Michalakes et al., 2001) and waves (either SWAN: Booj et al., 1999; or WaveWatch III; Tolman et al., 2014). Waves, in particular, play an important role in cohesive sediment dynamics through wave-enhanced bottom shear stresses, wave-induced near-bottom turbulence, and wave-induced nearshore circulation, but wave-induced fluid-mud layer processes are not represented."

*Specific comments:*

***Section 2.2.1: I'm not sure whether this is the best place to present fluxes into the bed. The alternative (which probably would be my preference) is to combine with erosion into a "bed water column exchange" section.***

This is a good suggestion, and we have re-arranged the paper to address it. We have added heading "2.3 Bed – Water Column Exchange" with subsection "2.3.1. Fluxes into the bed – Critical shear stress for deposition" (with the contents of previous Section 2.2.1) and a new subsection "2.3.2. Fluxes out of the bed – Resuspension" which includes the erosion rate equation.

Previous section "2.3.2. Changes in floc size distribution within the bed" has been moved up as Section 2.2.3. We thank the reviewer for helping make this section clearer and more readable.

***Figure 3a,b: It would be helpful to also have the temporal evolution of G shown. Since the authors include the modelling results of Verney et al. (2011), it would be useful to explain the reason for the different model results during the first aggregation stage (initial distribution), instead of relying on the (initial distribution), instead of relying on the reader checking in Verney et al. (2011).***

We agree. We have added time-dependent curves for G to Fig 3 a in the revised manuscript.

***Section 2.6: The new modules are added to the existing sediment transport model in ROMS (Warner et al., 2008) and in COAWST, which includes waves. The presence of bedforms and waves may induce pressure gradients at the sediment bed, which would in turn induce interstitial porewater flow in the bed. This process can entrain fine particles into a coarser sediment bed (e.g., Huettel et al., 1996, Limnol. Oceanogr.,41(2), 1996, 309-322). It would be welcome for the authors to comment on this process and its inclusion (or not) in the present framework.***

We agree this process might be important, especially for biogeochemical constituents. It is not represented in this version of ROMS because small scale bottom topography is not resolved and our version of ROMS is not yet coupled with a groundwater transport model, and we have not explored a sub-grid scale parameterization of this process. We have added to the discussion a short list of processes that are not addressed in the model, as follows: "However, not all of the processes associated with cohesive or mixed sediment have been included. For example, fluid muds and non-Newtonian flows are not represented (e.g., Mehta, 1991; 2014), nor is flow-induced infiltration of fine material into a porous bed (Huettel et al., 1999). Changes to the erodibility of mud that has been exposed at low tide (e.g., Paterson et al, 1990; Pilditch et al., 2008) or affected by flora or fauna (e.g., de Boer, 1981; de Deckere et al., 2001) are not considered."

***Figure 4: there appears to be a "kink" in the concentration for one specific profile (3560 microns?). What is the cause?***

The model solution becomes very sensitive, especially for larger particles, when both C and G are high, so it produces instabilities. We are not sure if these are real, or numerical artifacts, but they only occur under conditions with very high concentrations and turbulent shear.

***Figure 8: Caption should include details on what the different panels (a, b, c, d) show.***

We agree, and are not sure where those details went! We changed the caption to read as follows: "Figure 8. Comparison of estuarine turbidity maxima simulations with and without floc dynamics. a) Two-dimensional (along-estuary and vertical) snapshot of suspended particle concentrations (shaded)

without floc dynamics near the end of flood tide. b) Snapshot of suspended particle concentrations at the same time in the simulation, but with simulated floc dynamics (shading), overlain by contours of mean particle diameters. c) Along-estuary profiles of bed elevations for simulations without floc dynamics (red) and with floc dynamics (black) at the peak of flood tide (solid lines) and at post-flood slack tide (dashed lines). d) Along-estuary profiles of mean particle diameter in the top layer of the seabed, using the same notation as (c). The model was initialized with a uniform suspended-sediment concentration of 0.1 kg/m3 in the 37-μm class."

***Technical corrections:***

***Line 79: one too many that***

Fixed.

***Line 145-146 vs lines 115-116: Repetition, please remove one of the two.***

We modified the text near lines 145-146 to help address the Referees first comment, so there is no longer repetition.

Response to interactive comment by Anonymous Referee #3 on "Cohesive and mixed sediment in the Regional Ocean Modeling System (ROMS v3.6) implemented in the Coupled Ocean Atmosphere Wave Sediment-Transport Modeling System (COAWST r1179)" by Christopher R. Sherwood et al. Comment received 20 March 2018.

The authors thank Anonymous Referee #3 for comments on our manuscript. Here, we respond to those comments and indicate changes we have made in the manuscript to address them. Referee comments are reproduced in ***bold+italics***; our response is in plain text.

***OVERVIEW OF THE MS: This manuscript describes and demonstrates algorithms for treating fine and cohesive sediment that have been implemented in the Regional Ocean Modeling System (ROMS). These include: floc dynamics (aggregation and disaggregation in the water column); changes in floc characteristics in the seabed; erosion and deposition of cohesive and mixed (combination of cohesive and non-cohesive) sediment; and biodiffusive mixing of bed sediment. These routines supplement existing non-cohesive sediment modules, thereby increasing our ability to model fine-grained and mixed-sediment environments. Additionally, the manuscript describes changes to the sediment bed-layering scheme that improve the fidelity of the modeled stratigraphic record. Finally, the manuscript provides examples of these modules implemented in idealized test cases and a realistic application.***

————————————————————————————————————————

***MY REVIEW COMMENTS: I see these finding to be very interesting and of great importance, especially for coastal environmental management, where the accurate prediction of the movement and transport of both purely cohesive and mixed sediments is vital, for issues such as navigational waterways and water quality. The manuscript is generally well written and correctly structured, some relevant illustrations, and an appropriate range of relevant literature cited and referenced. The study aims and objectives are clearly defined on pp 4. However, the following points need to be addressed in detail, before this manuscript can be considered for publication.***

***Well written abstract. I would like to see a few more key quantitative findings reported there, in particular in terms of typical SSC levels and hydrodynamic ranges assessed by the model, plus some key model output values. I would also suggest doing the same for the Conclusion (pp30-31).***

This suggestion touches on an important question: over what range of conditions is the model applicable? Strictly speaking, the model applies to dilute suspensions at high Reynolds number (fully turbulent flow). SSC must be low enough that particle influences on turbulence dissipation can be neglected (Hsu et al., 2003), and certainly low enough that the flow is approximately inviscid Newtonian. We have not quantified the sediment concentrations or range of hydrodynamic parameters that ensure these conditions, but a common boundary for fluid mud (where viscoplastic properties become important) is 10 kg/m3 (Einstein and Krone, 1962; Kirby, 1988). We initialized runs with concentrations up to that limit to investigate equilibrium floc diameters (Section 3.1.2 and Fig. 3c). The units on Fig 5a are incorrect and have been corrected on the revised ms…these are integrated SS inventories over a depth of 20 m, and should have units of kg/m2…the maximum concentrations near the bed were about 5.4 kg/m3. Most of the simulations we presented were run with much lower concentrations of ~0.2 to 2 kg/m3 (Fig 3a,b; Fig 4; Fig 8).

Because we did not explicitly explore the range of model applicability, we would prefer not to quote numbers in the Abstract or Conclusion, but we have added text to the discussion to clarify the conditions under which the model should apply.

*In Section 2 – Model Processes: I would like to see a little more background on sediment transport process theory. This would assist the reader with fundamentals behind how the new model opporates.*

*In Section 2.2 – Floc Processes: again, I think this section would benefit by having some brief flocculation theory review presented before the floc model description.*

The main focus of the paper is to describe the modeling methods we have implemented. Source papers that can provide a more complete background have been added, and a paragraph providing more background on the floc model approach has been added to Section 2, as described below.

*I think it would be good to briefly outline the range of different approaches used in flocculation modeling, and why the approach used in this model was chosen.*

Good suggestion. We added a paragraph in Section 2 describing the difference between distribution-based and class-size-based models and a justification for our choice of a class-size-based approach. This paragraph also cites references to some of the classic papers for settling-velocity modeling, including Van Leussen (1998), Winterwerp (2006), Manning and Dyer (2007), Khelifa and Hill (2006) and Soulsby et al. (2013). I think this helps put our approach in context.

*Other aspects that I would like to see further updated in the manuscript, are slight updates with the Introduction section, where specific aspects could be further strengthened. I would like to recommend including some of the following references in the Introduction literature review. This would significantly strengthen the literature reviewed in the manuscript. These would provide links to recent research findings that would provide synergy and context for the research reported in this manuscript. It would be good if aspects of the following publications were included in the Discussion. These four publications provide additional insights into cohesive sediment flocculation and associated settling dynamics, together with applied modelling:*

*- Mehta, A.J., Manning, A.J. and Khare, Y.P. (2014). A Note on the Krone deposition equation and significance of floc aggregation. Marine Geology, 354, 34-39, doi.org/10.1016/j.margeo.2014.04.002.*

We have added a sentence citing this paper in Section 1.2

*- Mietta, F., Chassagne, C., Manning, A.J. and Winterwerp, J.C. (2009). Influence of shear rate, organic matter content, pH and salinity on mud flocculation. Ocean Dynamics, 59, 751-763, doi: 10.1007/s10236-009- 0231-4.*

We have added these to the References and cited it in the section of the Discussion where we itemize processes that are not included in our model.

*- Soulsby, R.L., Manning, A.J., Spearman, J. and Whitehouse, R.J.S. (2013). Settling velocity and mass settling flux of flocculated estuarine sediments. Marine Geology, doi.org/10.1016/j.margeo.2013.04.006.*

This paper is cited on line 61.

*- Winterwerp, J.C., Manning, A.J., Martens, C., de Mulder, T., and Vanlede, J. (2006). A heuristic formula for turbulence induced flocculation of cohesive sediment. Estuarine, Coastal and Shelf Science, 68, 195-207.*

This paper is cited on line 38.

*These two publications have demonstrated the importance of biological cohesion on bed sediments, as this has an important role on erosion threshold and bio-stability:*

*-Malarkey, J., Baas, J.H., Hope, J.A., Aspden, R.J., Parsons, D.R., Peakall, J., Paterson, D.M., Schindler, R.J., Ye, L., Lichtman, I.D., Bass, S.J., Davies, A.G., Manning, A.J., Thorne, P.D. (2015). The pervasive role of biological cohesion in bedform development. Nature Communications, DOI: 10.1038/ncomms7257*

*. - Parsons, D.R., Schindler, R.J., Hope, J.A., Malarkey, J., Baas, J.H., Peakall, J., Manning, A.J., Ye, L., Simmons, S., Paterson, D.M., Aspden, R.J., Bass, S.J., Davies, A.G., Lichtman, I.D. and Thorne, P.D. (2016). The role of biophysical cohesion on subaqueous bed form size. Geophysical Research Letters, 43, doi:10.1002/2016GL067667.*

These papers deal with biological cohesion of normally non-cohesive sediment. This is a process that is not addressed by our model. We have added these references and cited them in the section of the Discussion where we itemize processes that are not included in the model.

*This publication provides good general overviews of cohesive sediment dynamics:*

*-Mehta, A.J. (2014). An Introduction to Hydraulics of Fine Sediment Transport, World*

*Scientific, Hackensack, N. J.*

This book is cited on line 33 and elsewhere in the manuscript.

*Although the manuscript mentions mixed sediments in Section 2.5, it reports very little about the effects of mixed sediment flocculation. As much of the model application could be utilized in areas where there are sand / silt / clay, and biological cohesions, the manuscript would benefit from the citation of some of these recent key publications on the flocculation processes of cohesive and mixed fine-grained sediment suspension, as these outline key processes relating to these suspended sediment types:*

*\* Manning, A.J., Baugh, J.V., Spearman, J.R., Pidduck, E.L. and Whitehouse, R.J.S. (2011). The settling dynamics of flocculating mud:sand mixtures: Part 1 – Empirical algorithm development. Ocean Dynamics, INTERCOH 2009 special issue, doi:*

*10.1007/s10236-011-0394-7.*

*\* Manning, A.J., Baugh, J.V., Spearman, J. and Whitehouse, R.J.S. (2010). Flocculation Settling Characteristics of Mud:Sand Mixtures. Ocean Dynamics, doi: 10.1007/s10236-009-0251-0.*

*\* Spearman, J.R., Manning, A.J. and Whitehouse, R.J.S. (2011). The settling dynamics of flocculating mud:sand mixtures: Part 2 – Numerical modelling. Ocean Dynamics, doi: 10.1007/s10236-011- 0385-8.*

We have added the following text to the Discussion: "It is important to note that the mass settling fluxes of mixed (sand + mud) suspensions may be overestimated if their interactions are not considered, as is the case in the approach taken here (Manning et al., 2010, Manning et al., 2011)." We also added two references to the citations (Spearman et al., 2011 was previously cited on line 63 of the manuscript).

*In terms of the erosion-depositional cycle, Spearman and Manning (2008) have demonstrated that the threshold shear stresses for both deposition and erosion can operate simultaneously, in order to correctly mass-balance accretion and erosion levels of cohesive sediments during tidal cycles in shallow water locations. I would like to see this commented on within the context of your own study findings.*

*- Spearman, J. and Manning, A.J. (2008). On the significance of mud transport algorithms for the modelling of intertidal flats. In: T. Kudusa, H. Yamanishi, J. Spearman and J.Z. Gailani, (Eds.), Sediment and Ecohydraulics - Proc. in Marine Science 9, Amsterdam: Elsevier, pp. 411-430, ISBN: 978-0-444-53184-1.*

This process is incorporated in the model and described in Section 2.2.1 (now 2.3.1). We have cited this paper in that section.

*I would like to see the Discussion (Section 5) expanded slightly, with some comparisons made with other commonly used sediment transport modeling approaches. Some quatification (also in a summary Table) to these comparisons would be helpful. This could advise the reader on where significant improvements and advances have been made with this new modeling approach. It would also be good to comment on the possible limitations on this new modeling approach.*

We have compared the model results for individual processes with results of others (e.g., flocculation and biodiffusion in this paper; bedload transport in Warner et al., 2008). We have touched on the significant improvements we feel this model offers. We think that quantitative comparison of our results with other models is beyond the scope of this paper, but we hope that future efforts may undertake this. We have not changed the manuscript to address this comment.

*In summary, I think these findings are significant and are worthy of publication in GMD.*

We thank the referee for providing input; we feel that this has helped us improve the paper.

[revised manuscript text omitted]

---

## Author Response (AR2)

Author technical corrections in response to the Topical Editor Decision "Cohesive and mixed sediment in the Regional Ocean Modeling System (ROMS v3.6) implemented in the Coupled Ocean Atmosphere Wave Sediment-Transport Modeling System (COAWST r1179 [now r1234])" by Christopher R. Sherwood et al.

We thank Topical Editor Guy Munhoven guidance through this process. His final technical remarks are shown here in **_bold+italics_**; our response is in normal text.

*Topical Editor Decision: Publish subject to technical corrections (11 Apr 2018) by Guy Munhoven*

*Comments to the Author:*

*Dear Chris,*

*Dear co-authors,*

*We have received three anonymous referee comments for your manuscript ``Cohesive and mixed sediment in the Regional Ocean Modeling System (ROMS v3.6) implemented in the Coupled Ocean Atmosphere Wave Sediment-Transport Modeling System (COAWST r1179)''.*

*All of the referees rate your manuscript "Good" or "Excellent" in all of the four rubrics (Scientific significance, Scientific quality, Scientific reproducibility, Presentation quality)*

*Anonymous Referees #1 and #2 recommend minor revision, while Anonymous Referee # 3 recommends major revision. Anonymous Referee #1 has already expressed his/her satisfaction regarding your reply to her/his comments.*

*You have posted comprehensive replies to the referees comments, and I find that all of the referees' questions, comments and recommendations have been adequately dealt with.*

*As Anonymous Referee #3 recommends major revision, I have taken more time to go over this referee's comments and your reply (Editor review). I find that anonymous Referee #3 essentially asked for the fundamentals to be better presented (more comprehensive literature review). The requested changes thus mostly fall into category "Scientific quality" in our synthetic assessment form. In that category, your manuscript was rated "Excellent" by Anonymous Referee # 3. I am therefore not sending out your revised manuscript out for review to the referee, but restrict this round to an Editor Review only, the more since the other two referees requested minor review only.*

*Upon examination, I find again that you have replied in an adequate manner to the referee's comments and amended the text accordingly.*

*I am pleased to inform you that I can now accept your manuscript for publication in Geoscientific Model Development, subject to technical corrections*

Thank you. We are honored that the paper has been accepted and look forward to publishing in GMD.

*There remain a few minor points to clarify or correct.*

*(1) In your reply to the points that Anonymous Referee #3 raised in RC4 (https://doi.org/10.5194/gmd-2017-267-RC4) in the paragraph on page C3 that starts with ``These four publications ...'', there is some confusion: you indicate that - Mehta et al. (2014) - the first of the four listed by the referee -is*

*now cited in a sentence added to Section 1.2. This does not happen to be the case. The paper by Mehta et al. (2014) is not cited at all.*

We have added the sentence "This has implications for deposition rates (Mehta et al., 2014)." to section 1.2, and we have added Mehta et al. (2014) to the References

*- Mietta et al. (2009) has been added to the Discussion where you itemize processes that are not included in the model. This does not happen to be the case either. Mietta et al. is cited in a new sentence in Section 1.2.*

*Could you please check this, so that the reply to the referee's comments is consistent with the text of the revised manuscript.*

We have added the following sentence to the discussion near line 695: "The floc model does not explicitly account for the effects of organic matter content, pH, or salinity on flocculation rate (e.g., Mietta et al., 2009); these influences are subsumed into user-adjustable parameters."

*(2) There is some section/subsection numbering mismatch in the revised manuscript: - 2.2 has 2.2.1 and 2.2.3 as subsections (but no 2.2.2) - 2.3 has 2.2.1 (sic) and 2.2.2 (sic) as subsections (instead of 2.3.x)*

*Please correct the numbering.*

Corrected.

*(3) Finally, please make sure that the information about SVN revisions in the manuscript will be consistently updated: at lines 4 (title), 17, 712 and 738 in the revised manuscript, where either 1179 or XXXX is indicated. Please also include the correct contribution number at line 735.*

We have updated the SVN revision number in three places, and in the Supplement. The correct version is 1234. We also added: "This paper is Contribution Number 3741 of the Virginia Institute of Marine Science, College of William & Mary", and corrected the CKH affiliation. We also added the postal code for JPR.

Other minor corrections include:

Spelling correction for "Acknowledgements"

Addition of a List of Figures.

Per the instructions in https://www.geoscientific-model-development.net/for_authors/manuscript_preparation.html, the following changes have been made:

Title and author information for the supplement has been removed.

Equations have been changed to Eq., and units with / (e.g., m/s) have been consistently converted to exponential format (e.g, m s$^{-1}$).

Some other minor changes to punctuation have been made. A marked up version is available if you would like to see the changes itemized.

We have made minor changes to the figures, mostly to correct the format of the units.

*Thank you for considering Geoscientific Model Development for the publication of your model developments.*

*Best regards,*

*Guy Munhoven*

We thank Guy and the GMD editors for sage advice and support during this process.